# Obtaining and Characterizing Composite Biomaterials of Animal Resources with Potential Applications in Regenerative Medicine

**DOI:** 10.3390/polym14173544

**Published:** 2022-08-29

**Authors:** Narcisa Babeanu, Nicoleta Radu, Cristina-Emanuela Enascuta, Elvira Alexandrescu, Mihaela Ganciarov, Mohammed Shaymaa Omar Mohammed, Ioana Raluca Suica-Bunghez, Raluca Senin, Magdalina Ursu, Marinela Bostan

**Affiliations:** 1Faculty of Biotechnology, University of Agronomic Sciences and Veterinary Medicine of Bucharest, 59 Marasti Boulevard, 011464 Bucharest, Romania; 2Biotechnology, Bioresources and Analysis Departments, National Institute of Chemistry and Petrochemistry R&D of Bucharest, 202 Splaiul Independentei Street, 060021 Bucharest, Romania; 3Center of Immunology, Institute of Virology Stefan S. Nicolau, 285 Mihai Bravu Avenue, 030304 Bucharest, Romania

**Keywords:** composite biomaterials, collagen, chitosan, clotrimazole, antimicrobial

## Abstract

Raw materials, such as collagen and chitosan, obtained from by-products from the food industry (beef hides and crustacean exoskeletons), can be used to obtain collagen–chitosan composite biomaterials, with potential applications in regenerative medicine. Functionalization of these composite biomaterials is a possibility, thus, resulting in a molecule with potential applications in regenerative medicine, namely clotrimazole (a molecule with antibacterial, antifungal, and antitumor activity), at a mass ratio (collagen–chitosan–clotrimazole) of 1:1:0.1. This functionalized composite biomaterial has great potential for application in regenerative medicine, due to the following properties: (1) it is porous, and the pores formed are interconnected, due to the use of a mass ratio between collagen and chitosan of 1:1; (2) the size of the formed pores is between 500–50 μm; (3) between collagen and chitosan, hydrogen bonds are formed, which ensure the unity of composite biomaterial; (4) the functionalized bio-composite exhibits in vitro antimicrobial activity for *Candida albicans*, *Staphylococcus aureus*, and *Staphylococcus aureus* MRSA; for the latter microorganism, the antimicrobial activity is equivalent to that of the antibiotic Minocycline; (5) the proliferation tests performed on a standardized line of normal human cells with simple or composite materials obtained by lyophilization do not show cytotoxicity in the concentration range studied (10–500) μg/mL.

## 1. Introduction

In the last decade, studies carried out with biomaterials obtained from natural polymers have intensified, due to the multitude of natural resources from which they can be obtained (terrestrial species, marine species, micromycetes, and macromycetes) and due to their applications. Among the natural biopolymers of interest, we can find the following: (1) hydrolyzed collagen (it contains small peptides with a low molecular weight of 0.3–8 kDa, enriched in amino acids, and obtained by hydrolyzing native collagen from bones, skin, or connective tissue derived from animals or fish); (2) chitosan (obtained from the chitin exoskeletons of marine species, from insects or macromycetes and micromycetes); and (3) microbial cellulose (obtained by microbial biosynthesis) [1,2]. The most interesting biopolymers for the present work are represented by hydrolyzed collagen and chitosan, because the raw materials from which these can be obtained are in fact sub-products from the food or marine industries. It has been demonstrated that collagen peptides can be absorbed at the dermis level, stimulating the proliferation of fibroblasts, increasing the production of hyaluronic acid and activating the protection mechanisms against UVA radiation [1]. Regarding the types of amino acids from hydrolyzed collagen, the studies carried out showed that amino acids, such as glycine, proline, and hydroxyproline, are predominant in this biomaterial [2]. Hydrolyzed collagen has been used in recent years in clinical applications as a biomaterial in regenerative medicine (tissue regeneration); alone, this raw material cannot form three-dimensional networks due to the low molecular weight of the peptides in its composition, but it can be mixed with other natural biopolymers, such as bio-cellulose or chitosan, with which it is biocompatible, thus, forming hydrogen bonds. These mixtures, by cross-linking followed by lyophilization, lead to the obtainment of biocomposites with three-dimensional structures (sponges, biofilms), which are biocompatible, and which possess antibacterial activity [3,4].

Chitosan, another biomaterial of interest, a deacetylated polyaminosaccharide consisting of two monomers of D-glucosamine (deacetylated units) and N-acetyl-D-glucosamine (acetylated units) joined by β-(1,4) glycosidic bonds (3), is obtained by alkaline hydrolysis of chitin [5].

Chitin is found in nature in the following three forms: α-chitin, β-chitin, and γ-chitin, which differ between from one another by the position of their functional groups, as in Figure 1 [5]. The most abundant is α-chitin, which can be found in the cell walls of fungi and yeasts, and in the exoskeletons of crustaceans. The β-chitin is found in the squid endoskeleton and in giant tubes synthesized by *Pogonophora and Vestimentifera worms* [6].

Two antiparallel molecules per cell unit exist in α-chitin, but only one exists in β-chitin; in these structures, the networks are organized in foil that is kept together by hydrogen bonds. In α-chitin, hydrogen bonds do not allow diffusion of the small molecules in the crystalline phase while, in β chitin, hydrogen bonds between foils do not exist. As a result, the reactivity of β-chitin is greater than that of α-chitin [6].

Chitosan with a low average molecular mass tends to adopt a linear form (linear chains). These relaxing configurations expose the protonated amino group from the polymer chain, which can interact easily with target organisms. If the average molecular mass of chitosan increases to more than 600 atoms in the structure, then the chain tends to adopt a coiled structure. If the average molecular mass of chitosan increases even more, then the protonated amino groups are wrapped even more and the polymer loses its ability to interact with the target organisms [7]. Morin-Crini and collaborators explain chitosan versatility via its behavior at different pH conditions. In aqueous solutions with pH < pKa (pKa: 6.3–6.5), the amino groups from chitosan are protonated, and chitosan has a polycation behavior. At pH > pKa, the amino groups lose a proton and chitosan becomes more reactive and is more suitable for derivatizing [8]. Chitosan biocompatibility depends on the deacetylation degree; its antimicrobial properties depend on the average molecular mass, while its antioxidant properties are dependent on the deacetylation degree and average molecular mass [9]. Lopez et al. calculated the degree of deacetylation (*DDA*) of chitosan and/or chitosan oligomers with the following equation [10]:(1)$$DDA=97.67-(\frac{26.486\times A1665}{A3450})$$
where *A*1665 represents the absorbance of chitosan FTIR spectra at 1665 cm^−1^, and *A*3450 represent the absorbance of chitosan FTIR spectra at 3450 cm^−1^ [10].

Medical applications are due to their structure, which is similar to glycosaminoglycans existing in the extracellular structure of bones [11]. Antimicrobial activities for microorganisms, such as *Staphylococcus aureus (S. aureus*), *Candida albicans* (*C. albicans*), and *Escherichia coli* (*E. coli*), have been reported for chitosan [11]. Composite biomaterials made with bio-cellulose, collagen, and chitosan show local antimicrobial effects for *S. aureus* and *E. coli* [12].

The antibacterial activity of chitosan is given by the interaction between the amino groups (-NH_2_) of chitosan (which exist in solutions as positively charged polycation), with the microorganisms’ cell wall, which is negatively charged, causing its destruction and the loss of cell integrity. This is because (-NH_2_) groups possess a pair of non-participating electrons, that are protonated in media with pH < 6.3. Another mechanism of action involves the inhibition of enzymatic activity, due to the interactions between chitosan and DNA, which have the final effect of modifying the synthesis of messenger RNA. Other studies have demonstrated the following: (1) in the case of gram-positive microorganisms, the chitosan prevents cell division by forming non-covalent bonds of teichoic acid in the peptidoglycan layer of the cell wall; (2) in the case of gram-negative microorganisms, chitosan disrupts the absorption of nutrients, due to its electrostatic interactions with the anionic lipopolysaccharide groups existing on the surface of the microbial cell wall [12]. The number of protonated amino groups (-NH_2_) involved in antimicrobial activity is directly proportional with the deacetylation degree of the chitosan [4,13]. Younes and Rinaudo [6] report that the chitosan molecules with low average molecular mass penetrate the cell wall of bacteria, combining with their DNA, inhibiting mRNA synthesis and DNA transcription. In the case of gram-positive bacteria, antimicrobial activity is intensified with the increase in the average molecular mass of chitosan molecules, while the antimicrobial properties against gram-negative microorganisms demonstrate an inversely proportional increase to the molecular mass of chitosan molecules (the low molecular mass of chitosan molecules causes its high antimicrobial activity against gram-negative bacteria) [6]. Regarding antifungal activities, the same authors [6] report that chitosan reduces infections with phytopathogens, such as *Fusarium oxysporum*, *Botrytis cinerea*, *Phytophthora infestans*, or *Colletotrichum* sp. (anthracnose). Regarding the mechanism of action against fungi, it is supposed that the molecules of chitosan form a permeable film at the interface between fungal cells and the environment; this film has the following two functions: (1) interfering with fungal growth and (2) activating defense mechanisms in plants, such as chitinase accumulation and the synthesis of proteinase inhibitors.

Due to its biocompatibility, scaffolds of chitosan and the chitosan thin films are used in the treatment of patients with deep burns, being a potential biomaterial for repairing nerve lesions, blood vessels, and damaged bones, or as an extracellular matrix for hepatocytes [5,9].

The hydrogels of chitosan can be functionalized with different nanoparticles using gallic acid and sodium tripolyphosphate as cross-linking reagents [11]; by introducing nanoparticles of Cu [12], ZnO, or Ag in these hydrogels, inorganic–organic composites with properties of “drug delivery carriers” are obtained [13]. Hydrogels of chitosan which contain silver have antibacterial and regenerative properties for tissues (the chitosan presence accelerates wound healing). The action mechanism for hydrogels of chitosan with silver consists of cell wall disruption and DNA binding, thus, preventing bacterial replication. Commercial products based on chitosan hydrogels are as follows: Beschitin, Tegaderm, Trauma DEX, Talymed, Chitodine, and the Syvech Patch [5]. More anionic polysaccharides, such as proteins, collagen, and alginate, can form complex polyelectrolytes with chitosan hydrogels [13]. Glutaraldehyde, sodium alginate, genipin, or polyethylene glycol are used as cross-linking agents.

Magnetic gels can be obtained by combining chitosan’s hydrogels with magnetic nanoparticles, such as Fe_3_O_4_, γ-Fe_2_O_3_, and CoFe_2_O_4_ [13], with applications in medical imaging.

This biopolymer (the chitosan) is currently used in drug formulation, such as microspheres of chitosan with diclofenac, aspirin, 5-fluorouracil; in gels/nanogels with caffeine or lidocaine; in tablets with diclofenac and salicylic acid; in capsules/microcapsules with insulin; in thin films with testosterone; in sponges with triamcinolone acetonide; and in tablets of oxyphenbutazone coated with chitosan. Usually, chitosan is used as an excipient (filler agent), as an encapsulation reagent of sensible drugs, and as a carrier and supplier of medicine, proteins, peptides, antibiotics, growth factors, vaccines, DNA, and RNA [8,14].

The composites’ biomaterials, which contain chitosan, hydroxyapatite, and/or carbon nanotubes, [15,16] have potential applications in regenerative medicine due to their properties in promoting bone osteogenesis, while the sponges based on chitosan and alginate favors the differentiation of multipotent stromal cells [17]. Composite materials with collagen and chitosan or the biomimetic magnetic structures covered by composites that contain collagen, chitosan, hydroxyapatite, and magnetic particles, such as Fe_2_O_3_ or CoFe_2_O_4_, have promising medical applications [18,19,20], due to their ability to enhance the regeneration of damaged tissues or bones, with or without the influence of a magnetic field [18,19,20]. Different solid magnetic composites were developed from chitosan, collagen, cellulose, and super magnetic nanoparticles of Fe_2_O_3_, CoFe_2_O_4_, NiZnFe_2_O_4_, MnxZn_(1-x)_Fe_2_O_4_, and Ba_12_Fe_28_Ti_15_O_84_ (named SPION), or from hydrogels of collagen, chitosan, and superparamagnetic nanoparticles of Fe_2_O_3_ reticulated with genipin, with applications in wireless telecommunications, made over short distances [21,22].

Materials, such as collagen, gelatin, silk fibrin, hyaluronic acid, chitosan, alginate, hydroxyapatite, or bio-cellulose, combined with bioactive glass particles and/or different metallic cations, such as Sr^2+^, Mg^2+^, Zn^2+^, Ga^3+^, and Cu^2+^, are used to obtain composite biomaterials with raised mechanical resistance, with a role in enhanced tissue regeneration [23].

Another composite biomaterial developed from collagen, chitosan, and hydroxyapatite doped with magnesium has the ability to enhance wound healing, according to test results obtained from lab animals [24,25]. 

Other composite biomaterials obtained from chitosan and collagen at different mass ratios (80:20; 50:50; 20:80), reticulated with a glyoxal solution (5%, *w*/*w*), which have good mechanical resistance and porous structure with interconnected pores, are non-cytotoxic for human keratinocytes type NHEK and dermal fibroblast (NHDF), and these have potential applications in regenerative medicine, due to wound healing acceleration properties [26]. Other scientists developed biomaterials with collagen and essential oils or chitosan, essential oils, and polyvinyl alcohol (with essential oil concentrations in biomaterials of 2%) of *Melisa officinalis* or *Melaleuca alternifolia*, with antimicrobial activities for *S. aureus*, *E. coli*, or *C. albicans* [27,28]. Human recombinant collagen and chitosan modified with methacrylic anhydride can be used to imprint biocompatible 3D structures; these imprinted structures do not exhibit cytotoxic activity for HUVEC cell lines after 72 h [29]. Thin films obtained from chitosan, collagen, hydroxyapatite, and silver nanoparticles promote osteoblast proliferation and exhibit local antimicrobial activities against *S. aureus*, *E. coli*, and *Pseudomonas aeruginosa (P. aeruginosa)*, with an inhibition diameter of 2.5–3.5 mm [30].

Composite biomaterials based on chitosan, gelatin, and ciprofloxacin, cross-linked with genipin [31], exhibit antimicrobial activity for microorganisms, such as *E. coli*, *P. aeruginosa*, and *S. aureus*. In addition, preclinical tests performed on lab animals with these composites have shown that when these are applied on skin lesions, they speed up the healing process, down-regulate the level of pro-inflammatory cytokines, such as TNF-α, IL-6, and IL-1β, and up-regulate the level of anti-inflammatory cytokines, such as TGF-β1 [31]. 

Sionkowska et al. obtained composite thin films, with collagen, chitosan, and silver, with antimicrobial action against *S. aureus* [32]. The composite biomaterials based on collagen, chitosan, resveratrol, and doxycycline obtained by Tallapaneni and collaborators, for use in healing the wounds of diabetic patients, inhibit the development of pathogenic microorganisms, such as *S. aureus*, *S. aureus* MRSA, *P. aeruginosa*, and *E. coli*, (inhibition diameters obtained were more than 4 cm). Cytotoxicity tests performed with composite biomaterial concentrations in the culture medium of (31 ÷ 250) μg/mL), on Balb/3T3 murine fibroblast cell lines showed that, after 72 h, the cell viability was under 70% [33].

Other researchers have obtained collagen–chitosan–ciprofloxacin composite biomaterials [34] or ciprofloxacin collagen [35] with antimicrobial activities for *Escherichia coli* or *Staphylococcus aureus* [34,35]). Tests carried out on HUVEC cell lines [35] or murine fibroblasts demonstrated that this type of biomaterial had low cytotoxicity. Most of the studies carried out so far have shown that composite biomaterials made with collagen, chitosan, bio-cellulose, antioxidant compounds, and/or antibiotics exhibit superior mechanical stability and significant antimicrobial effects [36,37] and have potential applications in the wound treatment of diabetic patients. Composites made with chitosan and triclosan (an antiseptic reagent used for surgical sutures, implants, and medical devices) have antimicrobial activity and are effective and safe if the concentration of triclosan in composites does not exceed 0.3% [38]. Biomaterials, such as the simple chitosan or the composite biomaterials of the chitosan–collagen type, have the potential to be used to increase the biocompatibility of some metal alloys, developed for use in the form of implants, such as (1) metal composites based on Zn, Nb, Zr, and Co reinforced with magnesium or (2) metal composites containing Zn, Cu, Ti, Ca, and P [39,40].

The studies carried out at the lab level, on the metal composite based on Zn, Cu, Ti, Ca, and P, have shown good resistance to corrosion. Additionally, the determinations made “in vitro” with this type of metallic composite showed that its presence favors the proliferation of a Vero cell line, the measured cell viability having values higher than 95% after 24, 48, and 72 h of exposure [40].

Regenerative medicine requests new biocompatible formulations with enhanced properties. These types of formulations are currently obtained from cheap raw materials, such as the sub-products resulting from the food industry, such as animal hides and crustacean exoskeletons (chitin). These sub-products represent the main natural resources for obtaining biopolymers, such as hydrolyzed collagen and chitosan. A promising molecule with the possibility to enhance the properties of the biomaterials obtained from chitosan and/or collagen is clotrimazole, an imidazole derivative used in the treatment of fungal infections.

The aim of this study was to obtain and characterize composite biomaterials based on collagen hydrolyzate, chitosan, and clotrimazole.

The main objectives of the research were as follows:(1)obtaining collagen–chitosan composite biomaterials in the absence or the presence of an antibiotic reagent, with potential use in regenerative medicine;(2)characterization of the morphology of the composite biomaterials obtained, compared to the biomaterials resulting from raw materials (collagen and chitosan);(3)highlighting the structural characteristics of the obtained biomaterials (highlighting the functional chemical groups, the interactions between molecules, and the triple helix structure from collagen in simple or composite biomaterials;(4)evaluation of the antimicrobial properties of the simple or composite biomaterials;(5)evaluation of the cytotoxicity of the obtained simple biomaterials/composite biomaterials;(6)selection of composite biomaterial(s) with potential applications in regenerative medicine.

The research methodology applied in this research work is presented in Figure 2.

## 2. Materials and Methods

### 2.1. Obtaining Collagen Composite Materials: Chitosan: Clotrimazole

#### 2.1.1. Source of Chitosan

The chitosan solution was obtained as follows: 1 g of chitosan was dissolved in 0.5 M acetic acid solution, before stirring at 200 rpm (magnetic stirrer, Heidolph Instruments GmbH Schwabach, Germany) for 2 h. Pharmaceutical chitosan, obtained from crabs, purchased from DVR Pharm (Brasov, Romania), was used in the experiments. The solution was left to mature for 24 h before use. The determinations made by ESI/MS (electrospray ionization coupled with mass spectrometry) on this solution, showed that, after ripening, the system contains species with molecular mass between 0.6–3.1 kDa (Figure 3). The determinations were made using a device type LC–MS/TOF 6224 (AGILENT Technology Houston, TX, USA). 

#### 2.1.2. Source of Collagen

A hydrolyzed collagen with 2% collagen, obtained from beef skin residues (beef hides), at the National Institute for Textile and Leather R&D of Bucharest, Romania (INCDTP Bucharest) [35] was used for obtaining composite biomaterials.

The hydrolyzed collagen characteristics were as follows: density = 8.5 g/cm^3^; pH = 8.5; Ca_total_ = 4.21 (*w*/*w*); amino acids content (*w*/*w*): glycine = 0.49%; proline = 0.41%; alanine = 0.28%; arginine = 0.23%; glutamic acid = 0.2; tyrosine = 0.19%; valine = 0.2%. The ESI/MS analysis of the collagen hydrolyzate showed that it contained species with molecular mass between 0.3–2.25 kDa (Figure 4).

#### 2.1.3. Functionalization Reagent

Clotrimazole powder was used as a functionalization reagent, with 98% purity (Merck, Darmstadt, Germany).

#### 2.1.4. Obtaining Composite Biomaterials with Collagen-Chitosan-Clotrimazole

After 24 h, in the chitosan solution made at Section 2.1.1 clotrimazole and collagen were added, under stirring at 200 rpm. Clotrimazole was first added into the chitosan solution under stirring, after which collagen hydrolyzate was added. The formulated mixtures were homogenized at 200 rpm for 30 min, after which 20 mL of each experimental variant was poured into silicone molds. These were frozen at −20 °C for 24 h. The solid composite biomaterials (sponges) were obtained by freeze-drying the frozen biomaterials in three stages, as follows: (1) freezing at −55 °C for 1 min, (2) drying at −55 °C for 15 h, and (3) final drying at −40 °C for 10 min. The sublimation process was carried out under a vacuum using a Labconco 77530 lyophilizer (Labconco Co., Kansas City, MO, USA).

In the final mixtures, the collagen, chitosan, and clotrimazole mass ratios were as follows: 1:0:0 (S1); 0:1:0 (S2); 1:1:0 (S3); 3:1:0(S4); 1:1:0.1 (S5); 1:1:0.5 (S6).

The system containing collagen, chitosan, and clotrimazole at a mass ratio of 1:1:0.5 (S6) led to the precipitation of the antibiotic reagent and, for this reason, this solution was not used to obtain biomaterials.

### 2.2. Scanning Electron Microscopy Analysis

The morphology of the composite biomaterials was obtained through electron microscopy studies, carried out with FEI Quanta 200 scanning electron microscope with ambient scanning (Thermo Fisher Scientific Inc., Waltham, MA, USA). Magnifications of over 100,000× were used, and the equipment was operated at a pressure under 1.3 Pa.

### 2.3. Infrared Spectra Analysis

The infrared spectra were recorded in the range 400 ÷ 400 cm^−1^, using an FT-IR GX Perkin Elmer spectrophotometer (Perkin Elmer Ltd., Beaconsfield, UK), equipped with an ATR device, a Dynascan interferometer, and a DTGS (deuterated triglycine sulfate) detector.

### 2.4. Antimicrobial Activities

Antibacterial and antifungal activities were determined “in vitro”, on Petri plates with a solid culture medium, using the Kirby–Bauer diffusion disc method. The microorganisms used were *Candida albicans* ATCC 10231, *Escherichia coli* ATCC 25922, *Pseudomonas aeruginosa* ATCC 13388, *Staphylococcus aureus* ATCC 25923, and *Staphylococcus aureus* MRSA ATCC 335921. Each microbial inoculum was prepared in physiological serum, corresponding to the standard 0.5 McFarland, using fresh cultures of each microorganism. The microbial inoculum was spread on the surface of each plate with the help of a sterile swab. The culture media used were as follows: Mueller–Hinton medium for bacteria, and PDA medium (potato dextrose agar) for *Candida albicans* [41,42]. After 20 min, a sample of the composite biomaterial was deposited on the surface of each inoculated Petri dish. The composite biomaterials were tested in the form of small squares with a side of 6 mm, sterilely cut, in a laminar flow hood (laminar hood type Faster Bio 48, Cornaredo, Italy). Petri plates prepared in this way were incubated for 18 h at 37 °C for bacteria, and 18 h at 25 °C for fungi, using an incubator Cole Palmer H2200-H-E (Vernon Hills, IL, USA).

The antimicrobial activity was evaluated by comparing the resulting inhibition diameters, with those obtained in the case of the use of diffusive discs impregnated with antibiotics. For this purpose, diffusive disks of 6 mm diameter, impregnated with antibiotics (BioRad Lab, Hercules, CA, USA) were used. The following antibiotics were used as reference: clotrimazole (CT 10.8 mg/mL; or CT 0.1 mg/mL, clotrimazole dissolved in polyethylene glycol); ticarcillin/clavulanate 75 μg/10 μg (TCC75/10); ampicillin 25 μg (AMP25); tobramycin 10 μg (TOB 10); imipenem 10 μg (IPM 10); gentamicin 10 μg (Gen 10); piperacillin/tazobactam 100 μg/10 μg (PIT100/10); amoxicillin 10 μg (AX10); chloramphenicol 30 μg (C30), minocycline (min 5 mg/mL or min 2.5 mg/mL solutions were made with minocycline, produced by Merck (Merck KGaA, Darmstadt, Germany), dissolved in 20% dimethyl sulfoxide).

Antimicrobial experiments were performed in triplicate and the results are presented as average values, ±standard deviation.

### 2.5. Cell Proliferation Assay

As normal cells, the HUVEC line cell (ATCC PCS 100-010) was used. For the tests, a microbiological hood with laminar flow and an Elisa EZ 400 Biochrom analyzer (Holliston, MA, USA) was used. The composite biomaterials with the same mass were introduced in the culture mediums of vascular cell basal medium (ATCC PCS–100030) with HUVEC viable cells. After 5 min, all biomaterials were totally dissolved, and the resulting solution was used to make serial dilutions from each biomaterial composite. Four concentrations of each composite biomaterial were tested, respectively, as follows: 500 μg/mL, 250 μg/mL, 50 μg/mL, and 10 μg/mL. These solutions were used as nutritive media for HUVEC cells. After 24 h, 48 h, and 72 h, the numbers of viable cells were quantified using the CellTiter 96^®^ AQueous One Solution Cell Proliferation Assay (PromegaCo., Madison, WI, USA). The test is based on the reduction of yellow MTS tetrazolium salt by the viable cells and the generation of colored formazan soluble in the culture medium [43]. The product was spectrophotometrically quantified by measuring the absorbance at λ = 490 nm (42) using a Dynex plate reader (Dynex Technologies-MRS). Results were expressed as mean values of three determinations ± standard deviation (SD). Untreated cells served as control and were considered to have a proliferation index (PI) equal to 1. The proliferation index (PI), [42] was calculated according to the following Formula (2):(2)$$\mathrm{PI}=\frac{\%\;of\;viable\;cells\;from\;treated\;cells}{\%\;of\;viable\;cells\;from\;untreated\;cells}$$

## 3. Results

### 3.1. Morphology Analysis

The studies related to the morphology of the resulting biomaterials showed that although the lyophilized products containing collagen (Figure 5a1–a3) do not contain networks, they do have rare pores, and the resulting pores have sizes of 1 mm (Figure 5a2) and 500 μm (Figure 5a3). The lyophilized products obtained with collagen cross-linked with acetic acid are stable only at temperatures below 5 °C; at room temperature, they melt in 30 min (as such, they are not stable at room temperature). In the case of lyophilizates obtained from chitosan cross-linked with acetic acid (chitosan sponges), the images obtained by electron microscopy (Figure 6b1–b6) show the existence of some lamellar structures (Figure 6b1–b3), with the lamellae arranged in parallel; the distances between the parallel lamellas were <200 μm (Figure 6b2,b3). The biomaterial with chitosan has a structure consisting of networks (Figure 6b4), with pores arranged along the parallel lamellae. Biomaterial obtained by lyophilization of the acetic chitosan solution appears to have inhomogeneities (Figure 6b6). The size of the pores in the parallel networks of chitosan was <100 μm.

The results obtained in the case of sponges of chitosan from this study are comparable to those reported by Ikeda et al. [44], which used chitosan with a high degree of deacetylation (85%) and molecular mass = 100 kDa cross-linked with 2% acetic acid and neutralized with NaOH up to pH 7.4. The resulting lyophilized sponges have a porous structure, with a pore size <100 μm. The images obtained by electron microscopy for the composite biomaterials obtained at a collagen–chitosan = 1:1 mass ratio, show that the sponge obtained has a porous structure (Figure 7c1–c3, the average pore sizes from its structure being between 500 and 50 μm, (Figure 7c2,c3), and those with a size of 50 μm being predominant. If the mass ratio between collagen and chitosan increases (mass ratio collagen–chitosan = 3:1), then biomaterials with rare pores are obtained (Fi-gure 8d1), with dimensions between 500 μm (Figure 8d2) and 200 μm (Figure 8d3).

The results obtained in this case are similar to those obtained by Horn et al. [45], who obtained composite biomaterials from chitosan and collagen; the used chitosan had a low degree of deacetylation (16%) and was obtained from a squid species called *Loligo* sp. (a species of squid, whose endoskeleton contains chitin crystallized in the beta form) [46]. The collagen hydrolysate which was used was obtained via the alkaline hydrolysis of pig skins, for 24–96 h. The analysis of the morphology of the composite biomaterials obtained by lyophilization with hydrolyzed collagen for 96 h and chitosan from squid, showed that the presence of chitosan has the effect of reducing the size of the pores obtained, with the average size of the composite pores being between 56–163 μm. The biomaterial functionalized with clotrimazole, which contains collagen, chitosan, and clotrimazole in the mass ratio = 1:1:0.1 and borrows the structure of the composite biomaterial obtained at the same mass ratio but without clotrimazole. The images obtained by electron microscopy indicate a porous structure (Figure 9e1–e3), with pore sizes between 500 and 50 μm (Figure 9e1–e3) and an irregular structure in section (Figure 9e2).

### 3.2. Infrared Spectra Analysis

The infrared spectra of the obtained bio-composites were analyzed compared to the spectra of the raw materials from which they were obtained; respectively, collagen hydrolyzate, chitosan, and clotrimazole (Figure 10). By analyzing the bands that appear in the collagen hydrolyzate, we can see the presence of lipids. The intense band that appears at 1454.13 cm^−1^. was moved to lower wavenumbers in the three composite biomaterials, respectively, at 1438,06 cm^−1^, 1446,88 cm^−1^, and 1450.49 cm^−1^. The intense band that appears at 1260.9 cm^−1^ (amide III; CH_2_ wagging vibration) confirms the presence of glycine and proline in the collagen hydrolyzate.

The ratio between the intensity of the band appearing at 1260.9 cm^−1^ and the band appearing at 1454.13 cm^−1^ = 1.33 indicates the presence of the helix structure of collagen in the hydrolyzed biomaterial [46,47]. The obtained results demonstrate that the important bands characteristic of C-O bonds are shifted at higher wave numbers, which shows the presence of hydrogen bonds [48].

Thus, the intense band specific to the stretching vibration of the C-O bond from chitosan that appears at 1026.1 cm^−1^, in the collagen–chitosan = 1:1 composite appears as a band of average intensity at 1029.03 cm^−1^ and in the composite collagen–chitosan = 1:3 it appears in the form of a medium intensity band at 1032.37 cm^−1^.

In the composite biomaterial collagen–chitosan–clotrimazole = 1:1:0.1, this band appears with weak intensity at 1033.94 cm^−1^. The presence of the antibiotic reagent (clotrimazole) characterized by the band of high intensity that appears at 764.68 cm^−1^ specific to aromatic C-H (aromatic C-H bending) is confirmed by the band of weak intensity that appears shifted at 767.85 cm^−1^ in the composite (collagen–chitosan–clotrimazole) = 1:1:0.1. The presence of hydrogen bonds is also demonstrated by moving at higher values of the bands specific to the C=O bond from non-acetylated chitosan, which appears at 1630.62 cm^−1^ (weak intensity band). Thus, in the composite biomaterial (collagen–chitosan) = 1:1, the band corresponding to the C=O bond appears at 1633.37 cm^−1^ (intense band), in the composite (collagen–chitosan) = 3:1 it appears at 1633.31 cm^−1^ (very intense band) and in the composite collagen–chitosan–clotrimazole = 1:1:0.1 it appears at 1639.23 cm^−1^ (very intense band). The bands corresponding to the vibration of the NH functional group, present in the amino acids from collagen and, respectively, from the structure of chitosan appear in all infrared spectra. All these functional groups are involved in the formation of hydrogen bonds even if the resulting bands are overlapped [49,50].

### 3.3. Antimicrobial Activity

Studies performed in vitro, on the pathogenic microorganism activity show the following:(1)Collagen sponges (S1) do not exhibit antimicrobial activities for either microorganism (Figure 11a1–a4);(2)The sponges of chitosan (S2) and composite biomaterials which contain only chitosan (S2), only collagen–chitosan 1:1 (S3), collagen–chitosan 3:1 (S4), and collagen–chitosan–clotrimazole 1:1:0.1 (S4) do not exhibit antimicrobial activities against *P. aeruginosa*;(3)Regarding *Candida albicans*, the best activities were obtained for composite biomaterials obtained in the presence of clotrimazole (S3), giving an inhibition diameter of 23.9 mm (Figure 11a1), whereas the biomaterials S2, S3, and S4 show poor antimicrobial activity (diameter of inhibition under 6.3 mm) on this fungus.

The same results are obtained in the case of *Escherichia coli* for which all diameters of inhibition are situated under 8.2 mm, a fact which suggests only a small local antimicrobial activity (Figure 11a2).

**Figure 11 polymers-14-03544-f011:**
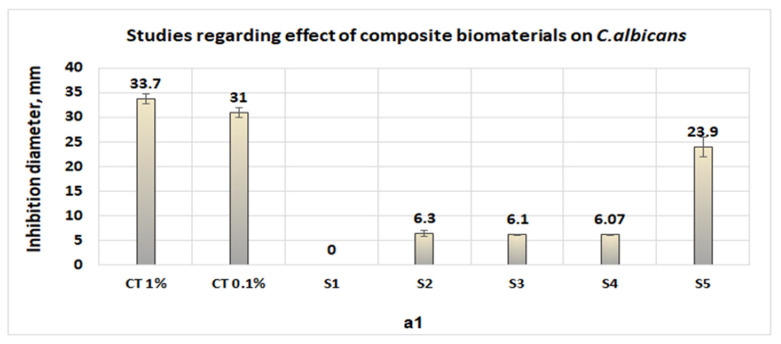
Studies regarding antimicrobial activities of biomaterials of the types S1 (collagen sponge), S2 (chitosan sponge), S3 (composite biomaterial obtained from collagen–chitosan at mass ratio 1:1), S4 (sponge obtained from collagen–chitosan mass ratio 3:1), and S5 (composite biomaterial obtained from collagen–chitosan–clotrimazole mass ratio = 1:1:0.1). (**a1**) antimicrobial activities on *Candida albicans*; (**a2**) antimicrobial activities on *Escherichia coli*; (**a3**) antimicrobial activities on *Staphylococcus aureus*; (**a4**) activities on *Staphylococcus aureus* MRSA.

In the case of *S. aureus*, biomaterials S2, S3, and S4 exhibit moderate antimicrobial activity (diameters of inhibition under 8.4 mm), but in the case of composite biomaterials which contain clotrimazole (S5), the diameter of inhibition obtained was 18.67 mm (Figure 11a3).

Regarding *S. aureus* MRSA, the behavior of S2, S3, and S4 were the same as in the case of *S. aureus* (the inhibition diameters obtained were under 8.2 mm), but in the case of composite biomaterial S5, the diameter of inhibition obtained (21.8 mm) is greater than the inhibition diameters obtained in the case of clotrimazole (0.1% or 1%) and tobramycin, being comparable with the inhibition diameter obtained in the case of minocycline 2.5 mg/mL (an inhibition diameter of 22 mm) (Figure 11a4).

The antimicrobial activity obtained against *S. aureus* and, respectively, against *S. aureus* MRSA is in accordance with the results obtained by other scientists [30,31,32,33,51,52], who obtained significant antimicrobial activity for *S. aureus*, *S. aureus* MRSA, and *P. aeruginosa* only in the case of composite biofilms which contain collagen, chitosan, and silver. Regarding the action mechanism for biomaterial composites with chitosan and clotrimazole, in the case of *S. aureus* or *S. aureus* MRSA, most probably chitosan binds to teichoic acids, increases membrane permeability, and causes membrane depolarization [53], acting synergically with clotrimazole.

### 3.4. Proliferation Studies

Proliferation studies, made with the five biomaterials (S1, S2, S3, S4, and S5 represent the same composite biomaterials as in Section 3.3.) at 24 h, 48 h, and 72 h revealed the following:-At 24 h, the presence of sponges with chitosan or collagen appears to stimulate the cell viability, so that at 500 μg/mL, in the case of S1, the cell viability attains 100% and, in the case of chitosan, it attains 109%. The cell viability increases proportionally with the concentration of S1 and S2 in the culture media. At the low concentrations of S1 and S2, cell viability attains 56.7% and 44%, respectively (Figure 12a1). The proliferation index obtained at the maximum concentration in the case of S1 and S2 attains a value = 1 and, respectively, 1.1, whereas at the low concentrations (10 μg/mL), the PI attains 0.6 (Figure 12a2). In the case of composite biomaterials with collagen and chitosan (biomaterials S3 and S4), cell viability attains the maximum value at 250 μg/mL, with 107.5% and 74.6%, respectively, with a corresponding PI of 1.1 and 0.7, respectively. At concentrations under 250 μg/mL, the values of cell viability are less than 64%. In the case of composite biomaterial S5, the maximum value for cell viability is obtained at 50 μg/mL, and the corresponding PI = 0.9; -After 48 h of exposure, cells appear to be stimulated by the tested biomaterials (Figure 13b1), except for the S4 for which, at 500 μg/mL, cell viability was 50% and PI = 0.5 (Figure 13b2). For the rest of the biomaterials, the cell viability was greater than 80% and PI > 0.8;-Measurements performed after 72 h of exposure (Figure 14c1,c2) show that the cell viability was greater than 93% for all samples except the S3 at c = 500 μg/mL, for which the viability decreased at 57.8%. This data confirm low cytotoxicity for samples tested and are in agreement with data reported by other scientists [35,44,46,47] for composite biomaterials types, such as collagen–biocellulose, chitosan–biocellulose, and collagen–chitosan–biocellulose, with or without the presence of the antibiotic reagent.

**Figure 12 polymers-14-03544-f012:**
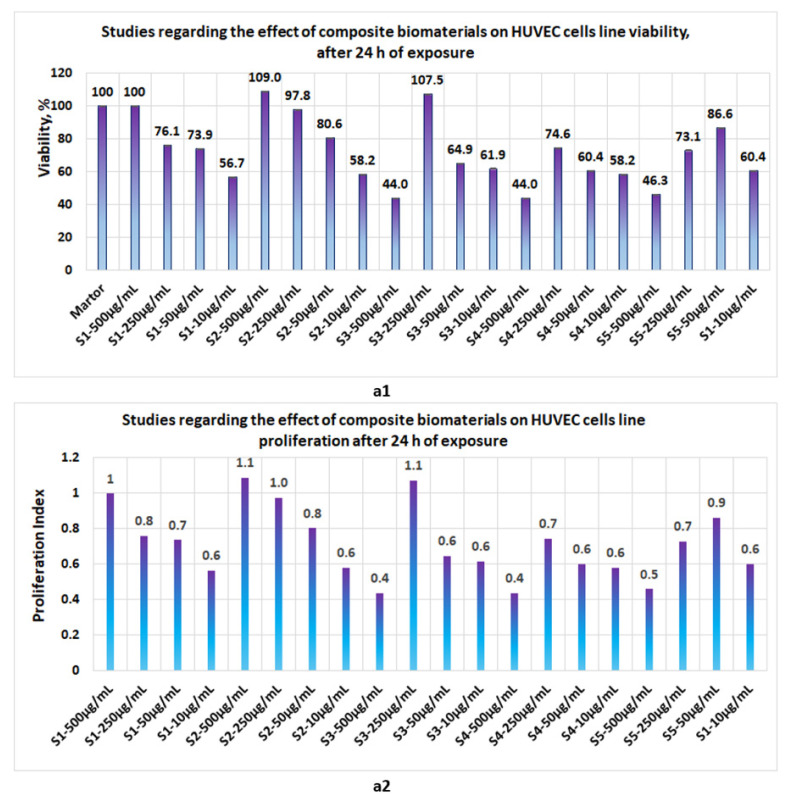
Studies regarding the effect of composite biomaterials on HUVEC cell lines after 24 h of exposure; (**a1**) influence on cell viability; (**a2**) influence on cell proliferation.

**Figure 13 polymers-14-03544-f013:**
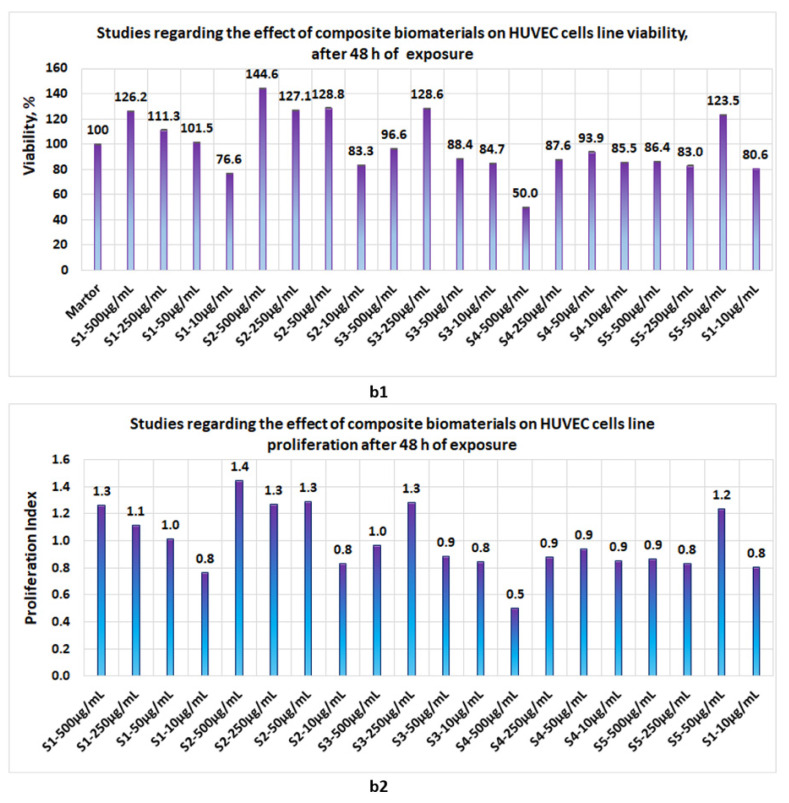
Studies regarding the effect of composite biomaterials on HUVEC cell lines after 48 h of exposure; (**b1**) influence on cell viability; (**b2**) influence on cell proliferation.

**Figure 14 polymers-14-03544-f014:**
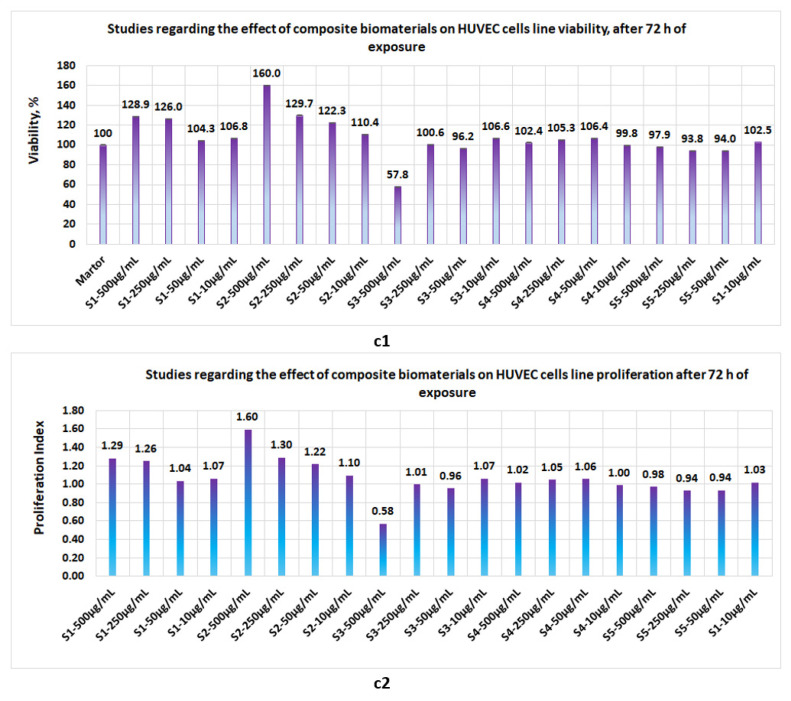
Studies regarding the effect of composite biomaterials on HUVEC cell lines after 72 h of exposure; (**c1**) influence on cell viability; (**c2**) influence on cell proliferation.

## 4. Discussion

The ESI/MS analysis showed that the collagen hydrolyzate contains species with a molecular mass of 0.3–2.2 KDa, values that are in agreement with reported data for this type of raw material [1,2]. The infrared analysis of the collagen hydrolyzate was used to highlight the existence of a structure of triple helix for collagen [46,47] in this solution. In the case of the chitosan solution made in 0.5M acetic acid, the existence of the monomers with an average mass of (0.645 ÷ 3.1) KDa was highlighted, which confirms the existence of chitosan structures of the “coil” or “wrapped coil” type in the acetic solution [7,10].

The physical–chemical analyses achieved showed that the biomaterials resulting from lyophilization have a porous structure, with the best composite biomaterial being obtained at a mass ratio between collagen and chitosan of 1:1. The composite biomaterial obtained at this mass ratio has a porous structure (pores with sizes between 1000 ÷ 100 μm) with interconnected pores, which ensures the circulation of oxygen through the biocomposite material [44,45,46]. In the biocomposite functionalized with clotrimazole, the average pore size decreases (the diameter of the resulting pores is 500 ÷ 50 μm), compared to the non-functionalized biocomposite, obtained at the same mass ratio.

The results obtained from the analysis of infrared spectra showed the following:In the collagen hydrolyzate used, collagen had the triple helix structure, the structure that is also preserved in the composite biomaterials obtained by lyophilization, for which the ratio between the two peaks is slightly shifted to the lower wave numbers, (the obtained values for the three composite biomaterials S4, S5 and S6 are as follows: 3, 2.14, and 1.5) [46,47];The theoretical degree of deacetylation of pharmaceutical chitosan calculated with Equation (1) is 65% [10]. Most probably, this degree of deacetylation represents the reason for inhomogeneities that appear in the morphology of the chitosan sponge (Figure 6b6). These inhomogeneities also appear in the morphology of the biomaterial composite of collagen–chitosan–clotrimazole, obtained at a mass ratio of 1:1:0.1 (Figure 9e2);In all of the three composite biomaterials, the presence of hydrogen bonds between the chitosan and collagen molecules was highlighted.

The tests carried out in vitro on the most common pathogenic microorganisms from medical practice showed that the biomaterials or composite biomaterials that do not contain clotrimazole have a local antimicrobial activity (inhibition diameters between 7–8 mm), results which are in agreement with other data reported in the literature [30,33].

The presence of clotrimazole significantly increases the antimicrobial activity of the biomaterial composite, for fungi, such as *Candida albicans* (for which clotrimazole is recognized to have selective action [54,55]) and, respectively, for gram-positive bacteria, such as *Staphylococcus aureus* and *Staphylococcus aureus* MRSA.

For medical applications, the biocomposite functionalized with clotrimazole has the greatest application potential in this field, as the existence of this type of composite biomaterial has not been reported in current literature. 

The tests performed ”in vitro” on *Staphylococcus aureus* with the composite biomaterial functionalized with clotrimazole showed that it has greater antimicrobial activity than the 0.1% clotrimazole solution, with the biological activity recorded as being comparable to that of the antibiotic tobramycin.

In the case of the tests carried out on *Staphylococcus aureus* MRSA, the results obtained showed that the antimicrobial activity of the functionalized composite biomaterial is higher than in the case of common antibiotics, such as clotrimazole (0.1 or 1%) and tobramycin, being comparable to the antibiotic minocycline, which is used in medical practice for treating and preventing infections with *Staphylococcus aureus* or *Staphylococcus aureus* MRSA [56]. Proliferation tests performed on a standardized human cell line (HUVEC) for 72 h showed that the obtained biomaterials are not cytotoxic, except for the composite biomaterial obtained at a mass ratio of collagen–chitosan = 1:1, for which cell viability drops to 50%, at a concentration in the culture medium of 500 μg/mL. This value is most likely due to experimental errors, because at 24 and 48 h the proliferation index is higher than 1. This fact once again confirms the lack of cytotoxicity of biomaterials of this type, as reported in the literature [26,29,34,35].

## 5. Conclusions

Three types of composite biomaterials were obtained from collagen, chitosan, and clotrimazole. Morphology studies performed by electronic microscopy revealed that the best structure was obtained in the case of a mass report between collagen and chitosan of 1:1. In this case, the composite biomaterial shows a porous structure with average dimensions of 500–100 μm. Composite biomaterials obtained from collagen, chitosan, and clotrimazole at a mass ratio of 1:1:0.1 exhibit a porous structure with average dimensions of 500–50 μm.

Infrared spectra analysis reveals the existence of hydrogen bonds between collagen and chitosan and, in the case of composite which contains an imidazole derivative, the presence of specific bands due to vibration of C-H from aromatic rings confirms its presence. The composite biomaterials which contain only collagen and chitosan exhibit mo-derate antimicrobial activities against *Escherichia coli*, *Candida albicans*, *Staphylococcus aureus*, and *Staphylococcus aureus* MRSA, whereas the composite biomaterial which contains an imidazole derivative shows significant antimicrobial activity against *Candida albicans*, *Staphylococcus aureus*, and *Staphylococcus aureus* MRSA.

Proliferation studies performed on the HUVEC cell line showed that, after 24 h of exposure, the cell viability for the main samples was situated between 44–104%. After 48 h of exposure, viability for the main samples was situated between 80–126%, whereas after 72 h of exposure, cell viability in the case of the main samples was situated between 93.8–128.9%.

The novelty of this study consists in obtaining a porous composite biomaterial with interconnected pores, functionalized with clotrimazole, with a pore size = (500 ÷ 50) μm, for which the studies performed in vitro showed the following:
(1)It exhibits significant antimicrobial activity for gram-positive microorganisms, such as *Staphylococcus aureus* and, respectively, for antibiotic-resistant microorganisms, such as *Staphylococcus aureus* MRSA;(2)It does not exhibit cytotoxicity;(3)It has a potential for application in regenerative medicine, as a result of its own components (collagen, chitosan, and clotrimazole) that support tissue regeneration; it inhibits the development of pathogenic microorganisms, such as *Candida albicans*, *Staphylococcus aureus*, or antibiotic-resistant microorganisms, such as *Staphylococcus aureus* MRSA.


To validate the applicative potential of this functionalized composite biomaterial, the following additional studies are needed:Testing the obtained composite biomaterials obtained on different tumor cell lines;Carrying out the preclinical tests on the lab animals in order to evaluate the antimicrobial activity “in vivo”;Carrying out preclinical tests on lab animals to confirm the absence of cytotoxicity “in vivo”.

## Figures and Tables

**Figure 1 polymers-14-03544-f001:**
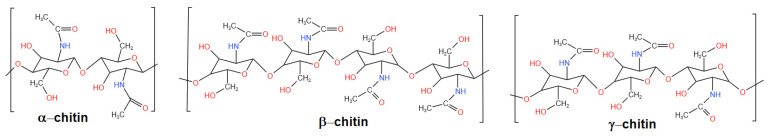
Allomorph form of chitin.

**Figure 2 polymers-14-03544-f002:**
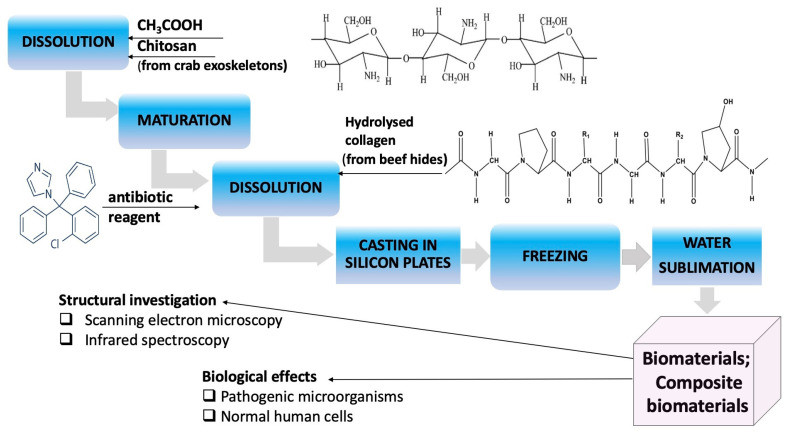
Research methodology applied in research studies.

**Figure 3 polymers-14-03544-f003:**
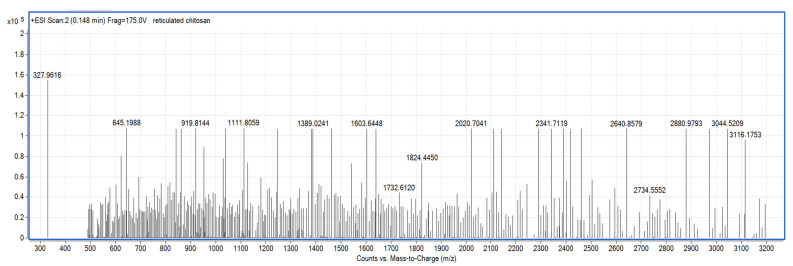
Analysis of chitosan solution reticulated with acetic acid after 24 h of maturation.

**Figure 4 polymers-14-03544-f004:**
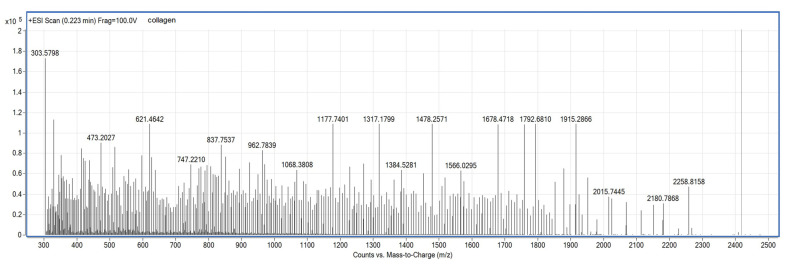
Analysis of collagen hydrolysate.

**Figure 5 polymers-14-03544-f005:**
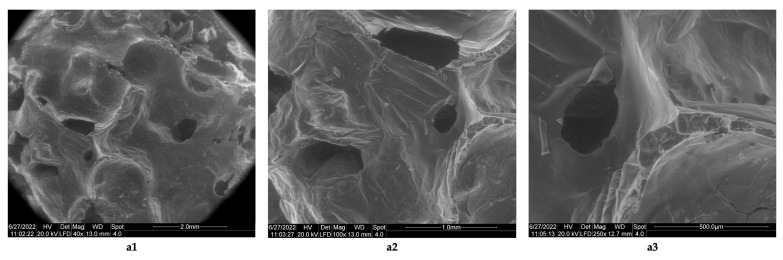
Morphology of collagen sponge obtained by lyophilization. (**a1**) Total view of collagen sponge, with rare pores (magnification 40×); (**a2**) view regarding unregulated pore geometry, most with diameters less than 1 mm (magnification 10×); (**a3**) dimensions of the rare small pore, with diameters of 250 microns (magnification 250×).

**Figure 6 polymers-14-03544-f006:**
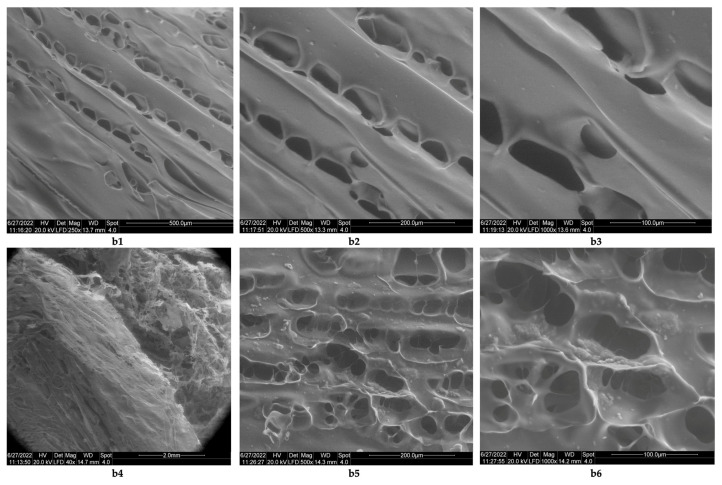
Morphology of chitosan sponge obtained by lyophilization: (**b1**) lamellar structure of chitosan sponge (magnification 250×); (**b2**) lamellas from a sponge of chitosan; distance between lamellas under 200 μm (magnification 500×); (**b3**) details regarding pores from the lamellar structure, with dimensions under 100 μm (magnification 1000×); (**b4**) interconnected pore distribution inside of sponge of chitosan (magnification 40×); (**b5**) pore geometry inside of chitosan sponge (magnification 500×); (**b6**) pore geometry from sponge of chitosan with inhomogeneities present in its internal structure (magnification 1000×).

**Figure 7 polymers-14-03544-f007:**
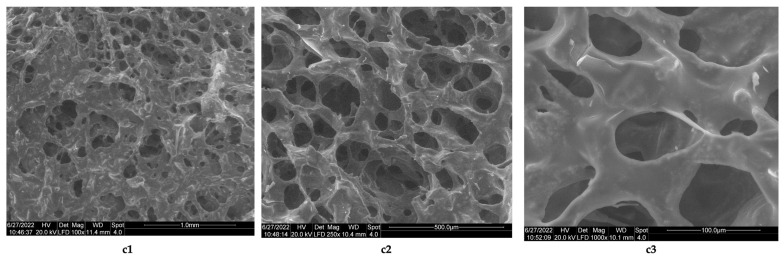
Morphology of composite biomaterial between collagen and chitosan, obtained at a collagen–chitosan mass ratio of 1:1 by lyophilization: (**c1**) porous structure of the collagen-chitosan sponge, with interconnected pores (magnification 100×); (**c2**) details regarding interconnected pore geometry (magnification 250×); (**c3**) pore geometry, with dimensions of 100 μm (magnification 1000×).

**Figure 8 polymers-14-03544-f008:**
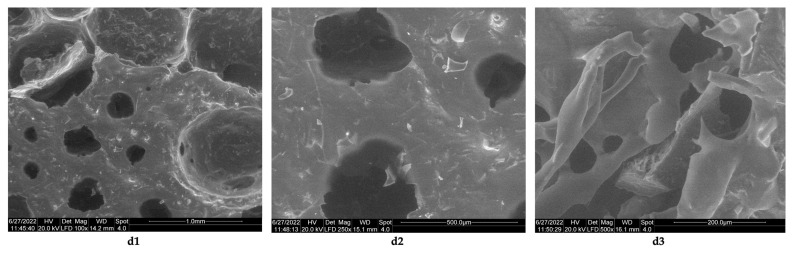
Morphology of composite biomaterial between collagen and chitosan obtained at a collagen–chitosan mass ratio of 3:1. (**d1**) porous structure of sponge with pores diameters under 1 mm (magnification 100×); (**d2**) pores geometry from the sponge with diameters under 500 μm (magnification 250×); (**d3**) small pores with dimension under 200 μm microns (magnification 500×).

**Figure 9 polymers-14-03544-f009:**
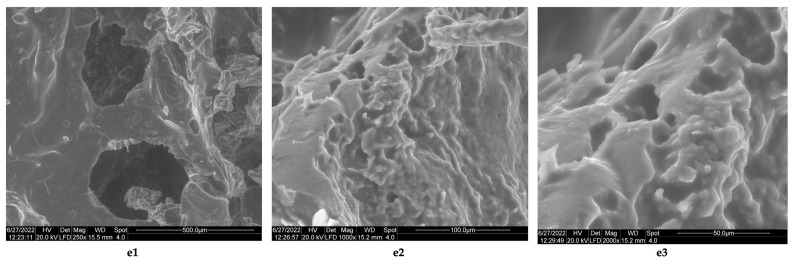
Morphology of sponge composite biomaterial obtained at collagen–chitosan–clotrimazole mass ratio of 1:1:0.1. (**e1**) porous structure of composite biomaterial, with pores diameter under 500 microns (magnification 250×); (**e2**) details regarding inhomogeneity of internal wall structures and presence of interconnected pores, with diameters under 100 μm (magnification 1000×); (**e3**) composite biomaterial with pores diameter under 50 μm (magnification 2000×).

**Figure 10 polymers-14-03544-f010:**
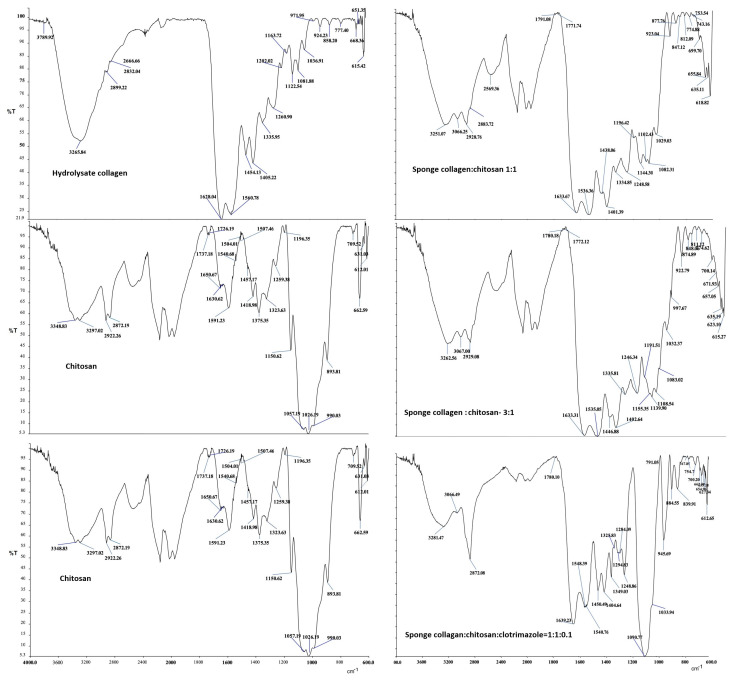
Infrared spectra of hydrolyzed collagen, chitosan, clotrimazole, and composite biomaterial with a collagen–chitosan mass ratio of 1:1; composite biomaterial with a collagen–chitosan mass ratio of 3:1; functionalized composite biomaterial with a collagen–chitosan –clotrimazole mass ratio of 1:1:0.1.

## Data Availability

The data presented in this study are available on request from the corresponding author.

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
