# Peer review of "Obtaining and Characterizing Composite Biomaterials of Animal Resources with Potential Applications in Regenerative Medicine"

_polymers, 2022, doi:10.3390/polym14173544_

Round 1

Reviewer 1 Report

The authors characterized composite biomaterials based on collagen hydrolizate, chitosan and clotrimazol. Studies performed on these composites showed a porous structure, with average dimensions of pores between (500-50) μm, the interactions between the two polymers being made by hydrogen bonding. All composite biomaterials exhibit moderate antimicrobial activity against Candida albicans, Escherichia coli, Staphylococcus aureus, and Staphylococcus aureus MRSA. Composite biomaterial obtained at a mass ratio of (collagen : chitosan : clotrimazole) = 1:1:0.1 exhibits significant antimicrobial activity against Candida albicans, Staphylococcus aureus, and Staphylococcus aureus MRSA. Proliferation studies performed in vitro on HUVEC cell lines, with composite biomaterials, collagen and chitosan sponge dissolved in culture media, at concentrations of (500-10) μg/mL reveals that after 48 and 72 h of exposure, cell proliferation was favored by the presence of all samples.

The paper may be published after major revision.

The authors need to interpret the meanings of the variables.

Please highlight your contributions in introduction.

The paper is well-written, I have to thank you to your effort.

Increase the font size of  figure1 and discuss the main features of that figure.

“The determinations made by ESI/MS measurements on chitosan maturated solution, (electrospray ionization coupled with mass spectrometry) (the device used: LC-MS/TOF 6224 AGILENT Technology Houston, USA) showed that after ripening, the chitosan solution contains species with a molecular mass between (0.32 - 3.1) kDa”, check this sentence.

Increase the font size of  figure 2  and discuss the main features of that figure.

What are the main features in SEM images in figure 3?

“The same results are obtained in the case of E. coli for which all diameters of inhibition are situated under 8.2 mm, a fact which suggests only a small local antimicrobial activity (Figure 5a2).”; discuss this sentence.

What are the main features in SEM images in figure 4?

Increase the font size of  figure 4  and discuss the main features of that figure.

The introduction should be supported by:  Biodegradable Magnesium Metal Matrix Composites for Biomedical Implants: Synthesis, Mechanical Performance, and Corrosion Behavior-A Review; Bistable morphing composites for energy-harvesting applications; In Vitro Degradability, Microstructural Evaluation, and Biocompatibility of Zn-Ti-Cu-Ca-P Alloy

Add more references from MDPI. To support introduction and discussion.

The abstract should be rewritten to reflect the significance of the proposed work. The current abstract shows a lot of background information.

Conclusion: What are the advantages and disadvantages of this study compared to the existing studies in this area?

The space between value and units may be eliminated.

The numbers for the all equations have to be provided.

Add future work in bullets.

Looking and wishes for the revised version.

Author Response

Dear Reviewer,

Thank you for your suggestion. All of these were solved as a follow

Request 1. Please highlight your contributions in introduction.

Answer in the revised article, as follow:

The aim of this study was to obtain and characterize composite biomaterials based on collagen hydrolyzate, chitosan and clotrimazole.

The main objectives of the research were:

1) obtaining collagen:chitosan composite biomaterials in the absence or the presence of an antibiotic reagent, with potential use in regenerative medicine;

2) characterization of the morphology of the composite biomaterials obtained, compared to the biomaterials resulting from raw materials (collagen, chitosan);

3) highlighting the structural characteristics of the obtained biomaterials (highlighting the functional chemical groups, the interactions between molecules, and the triple helix structure from collagen in simple or composite biomaterials ;

4) evaluation of the antimicrobial properties of the simple or composite biomaterials;

5) evaluation of the cytotoxicity of the obtained simple biomaterials/composite biomaterials;

6) selection of composite biomaterial(s) with potential applications in regenerative medicine.

The research methodology applied in this research work is presented in figure 2.

Request 2: a)Increase the font size of  the figure1 and b)discuss the main features of that figure.

Answer: a) All these were done in the manuscript.

  1. b) Discussions were introduced ( as a separate chapter), and this request was solved as follows:

The ESI-MS analysis showed that the collagen hydrolyzate contains species with a molecular mass of 0.3-2.2 KDa, values that are in agreement with reported data for this type of raw material [1-2]. The infrared analysis of the collagen hydrolyzate was used to highlight the existence of a structure of triple helix for collagen [46-47] in this solution. In the case of the chitosan solution made in 0.5M acetic acid, the existence of the monomers with an average mass of (0.645÷3.1) KDa was highlighted, which confirms the existence of chitosan structures type "coil" or ''wrapped coil'' in the acetic solution [7; 10].

Request 3.“The determinations made by ESI/MS measurements on chitosan maturated solution, (electrospray ionization coupled with mass spectrometry) (the device used: LC-MS/TOF 6224 AGILENT Technology Houston, USA) showed that after ripening, the chitosan solution contains species with a molecular mass between (0.32 - 3.1) kDa”, check this sentence.

Answer: The sentence was revised as follows:

The determinations made by ESI/MS (electrospray ionization coupled with mass spectrometry) on this solution, showed that, after ripening, the system contains species with molecular mass between (0.6 - 3.1) kDa (Figure 2.1). The determinations were made using a device type LC-MS/TOF 6224 AGILENT Technology Houston, USA).

Request 4. Increase the font size of figure 2  and discuss the main features of that figure.\

Answer: All these were done in the manuscript; discussion regarding figure 2 ( 2.1 in fact)

The ESI-MS analysis showed that the collagen hydrolyzate contains species with a molecular mass of 0.3-2.2 KDa, values that are in agreement with reported data for this type of raw material [1-2]. The infrared analysis of the collagen hydrolyzate was used to highlight the existence of a structure of triple helix for collagen [46-47] in this solution

Request 5. What are the main features in SEM images in figure 3?

The answer was presented in the Discussion chapter  of a manuscript as follows:

The physical-chemical analyses achieved showed that the biomaterials resulting from lyophilization have a porous structure, the best composite biomaterial being obtained at a mass ratio between collagen and chitosan of  1:1. The composite biomaterial obtained at this mass ratio has a porous structure (pores with sizes between (1000÷100)μm) with interconnected pores, which ensures the circulation of oxygen through the biocomposite material [44-46]. In the biocomposite functionalized with clotrimazole, the average pore size decreases (the diameter of the resulting pores is 500÷50 μm),  compared to the non-functionalized biocomposite, obtained at the same mass ratio.

Request 6. “The same results are obtained in the case of E. coli for which all diameters of inhibition are situated under 8.2 mm, a fact which suggests only a small local antimicrobial activity (Figure 5a2).”; discuss this sentence.

The answer was presented in the  Discussions chapter  of the manuscript, as follows:

               The tests carried out in vitro on the most common pathogenic microorganisms from medical practice showed that the biomaterials or composite biomaterials that do not contain clotrimazole have a local antimicrobial activity (inhibition diameters between 7-8 mm), results which are in agreement with other data reported in the literature [30; 33].

Request 6. What are the main features in SEM images in figure 4?

The answer was presented in the   manuscript (in figure 5 I think), was presented in the  Discussion chapter  of the manuscript as follows:

               The tests carried out in vitro on the most common pathogenic microorganisms from medical practice showed that the biomaterials or composite biomaterials that do not contain clotrimazole have a local antimicrobial activity (inhibition diameters between 7-8 mm), results which are in agreement with other data reported in the literature [30; 33].

               The presence of clotrimazole significantly increases the antimicrobial activity of the biomaterial composite, for fungi such as Candida albicans (for which clotrimazole is recognized to have selective action [54-55]) and, respectively, for gram-positive bacteria such as Staphylococcus aureus and Staphylococcus aureus MRSA.

               For medical applications, the biocomposite functionalized with clotrimazole has the greatest application potential in this field, as the existence of this type of composite biomaterial has not been reported in current literature. 

The tests performed ''in vitro'' on Staphylococcus aureus with the composite biomaterial functionalized with clotrimazole showed that it has greater antimicrobial activity than the 0.1% clotrimazole solution, the biological activity recorded being comparable to that of the antibiotic Tobramycin.

In the case of the tests carried out on Staphylococcus aureus MRSA, the results obtained showed that the antimicrobial activity of the functionalized composite biomaterial is higher than in the case of common antibiotics such as Clotrimazole (0.1 or 1%) and Tobramycin, being comparable to the antibiotic Minocycline, which is used in medical practice for treating/preventing infections with Staphylococcus aureus or Staphylococcus aureus MRSA [56].

Request 7.

The introduction should be supported by:  Biodegradable Magnesium Metal Matrix Composites for Biomedical Implants: Synthesis, Mechanical Performance, and Corrosion Behavior-A Review; Bistable morphing composites for energy-harvesting applications; In Vitro Degradability, Microstructural Evaluation, and Biocompatibility of Zn-Ti-Cu-Ca-P Alloy

These requests were introduced in the   manuscript, as follows:

The hydrogels of chitosan can be functionalized with different nanoparticles using gallic acid and sodium tripolyphosphate as crosslinking reagents [11]; by introducing nanoparticles of Cu [12], ZnO or Ag in these hydrogels, inorganic-organic composites with properties of ‘’drug delivery carriers’’ are obtained [13]. Hydrogels of chitosan which contain silver have antibacterial and regenerative properties for tissues (the chitosan presence accelerates wound healing). The action mechanism for hydrogels of chitosan with silver consists of cell wall disruption and DNA binding, thus preventing bacterial replication. Commercial products based on chitosan hydrogels are as follows: Beschitin, Tegaderm, Trauma DEX, Talymed, Chitodine and Syvech-Patch [5]. More anionic polysaccharides such as proteins, collagen and alginate can form complex polyelectrolytes with chitosan hydrogels [13]. As crosslinking reagents for these, glutaraldehyde, sodium alginate, ge-nipin or polyethyleneglycol are used.

Magnetic gels can be obtained by combining chitosan's hydrogels with magnetic nanoparticles such as Fe3O4, γ-Fe2O3, and CoFe2O4 [13], with applications in medical imaging.

This biopolymer (the chitosan) is currently used in drug formulation, like microspheres of chitosan with diclofenac, aspirin, 5-fluorouracil; gels/nanogels with caffeine or lidocaine; tablets with diclofenac and salicylic acid; capsules/microcapsules with insulin; thin films with testosterone; sponges with triamcinolone acetonide, tablets of oxyphenbutazone coated with chitosan.  Usually, chitosan is used as an excipient (filler agent), as an encapsulation reagent of sensible drugs, and as a carrier and supplier of medicine, proteins, peptides, antibiotics, growth factors, vaccines, DNA and RNA [8, 14 ].

The composites’ biomaterials which contain chitosan, hydroxyapatite and/or carbon nanotubes, [15-16] have potential applications in regenerative medicine due to their properties in promoting bone osteogenesis; while the sponges based on chitosan and alginate favors the differentiation of multipotent stromal cells [17]. Composite materials with collagen and chitosan or the biomimetic magnetic structures covered by composites that contain collagen, chitosan, hydroxyapatite and magnetic particles, such as Fe2O3, or CoFe2O4, have promising medical applications [18-20], due to their properties to enhance the regeneration of damaged tissues or bones, with or without the influence of the magnetic field [18-20]. Different solid magnetic composites were developed from chitosan, collagen, ce-llulose and super magnetic nanoparticles of Fe2O3, CoFe2O4, NiZnFe2O4, MnxZn(1-x)Fe2O4, and Ba12Fe28Ti15O84  (named SPION) or from hydrogels of collagen, chitosan and superparamagnetic nanoparticles of Fe2O3 reticulated with genipin, with applications in wireless telecommunications, made on short distances [21-22.].

Materials, such as collagen, gelatin, silk fibrin, hyaluronic acid, chitosan, alginate, hydroxyapatite or bio-cellulose, combined with bioactive glass particles and/or different metallic cations, such as Sr2+, Mg2+, Zn2+, Ga3+, Cu2+, are used to obtain composite biomaterials with raised mechanical resistance, with a role in enhanced tissue regeneration [23].

  Another composite biomaterial developed from collagen, chitosan and hydroxyapatite doped with magnesium has the property to enhance wound healing, according to test results obtained on lab animals [24-25].

Request 8. Add more references from MDPI. To support introduction and discussion.

Answer

The introduction Chapter was extended in the manuscript, and due to this fact, the number of references used in this manuscript was raised at 56, with more references from MDPI, as follow:

  1. Sibilla, S.; Godfrey, M.; Brewer, S.; Budh-Raja, A.; Genovese, L. An Overview of the Beneficial Effects of Hydrolysed Collagen as a Nutraceutical on Skin Properties: Scientific Background and Clinical Studies. The Open Nutraceuticals Journal, 2015, 8, 29-42.
  2. Masoud, M. Production of hydrolyzed bovine collagen with orange juice concentrate. Egyptian Journal of Agricultural Research, 2017, 95(3),1205-1217.
  3. León-López, A.; Morales-Peñaloza, A.; Martínez-Juárez, V.M.; Vargas-Torres, A.; Zeugolis, D.I.; Aguirre-Álvarez, G. Hydrolyzed Collagen—Sources and Applications. Molecules 2019, 24, 4031.
  4. Mitelut, A.C., Tanase, E.E.; Popa, V.I.; Popa, M.E. Sustainable alternative for food packaging: chitosan biopolymer - a review. AgroLife Scientific Journal , 2015, 4(2), 52-61
  5. Rufato, K. B.; Galdino, J. P.; Ody, K. S.; Pereira, A. G.; , Corradini, E.; Martins, A. F.; Paulino, A. T.; Fajardo, A. R.; Aouada, F. A.; Porta, F. A. L.; Muniz, A. F. R. E. C. Hydrogels Based on Chitosan and Chitosan Derivatives for Biomedical Applications. In L. Popa, M. V.  Ghica, & C. Dinu-Pîrvu (Eds.), Hydrogels - Smart Materials for Biomedical Applications. IntechOpen, 2018, https://doi.org/10.5772/intechopen.81811
  6. Younes, I.; Rinaudo, M. Chitin and chitosan preparation from marine sources. Structure, properties and applications. Marine drugs, 2015, 13(3), 1133–1174.
  7. Kou, S.(G); Peters, L.; Mucalo, M. Chitosan: A review of molecular structure, bioactivities and interactions with the human body and micro-organisms, Carbohydrate Polymers, 2022, 282, 119132.
  8. Morin-Crini, N.; Lichtfouse, E.; Torri, G.; Crini, G. Fundamentals and Applications of Chitosan. Sustainable Agriculture Reviews 35. Chitin and Chitosan: History, Fundamentals and Innovations, 35, Springer International Publishing AG, 2019, 338, 2019, Sustainable Agriculture Reviews.
  9. Islam, S.; Bhuiyan, M.A.R.; Islam, M.N. Chitin and Chitosan: Structure, Properties and Applications in Biomedical Engineering. J Polym Environ ,2017, 25, 854–866.
  10. Lopes, L.S; Michelon, M.; Duarte, L.G.R.; Prediger, P.; Cunha, R.L.; Picone, CSF. Effect of chitosan structure modification and complexation to whey protein isolate on oil/water interface stabilization,Chemical Engineering Science,2021, 230, 116124.
  11. Lamarra, J.; Damonte, L.; Rivero, S.; Pinotti, A Structural Insight into Chitosan Supports Functionalized with Nanoparticles. Advances in Materials Science and Engineering, 2018, 1-11,
  12. Nie, J.; Wang, Z; Hu, Q. Chitosan Hydrogel Structure Modulated by Metal Ions. Sci Rep., 2016, 6, 36005
  13. Fu, J.; Yang, F.; Guo,Z. The chitosan hydrogels: From structure 1 to function New J. Chem., 2018, 1-43.
  14. Iacob, A.T.; Lupascu, F.G.; Apotrosoaei, M.; Vasincu, I.M.; Tauser, R.G.; Lupascu, D.; Giusca, S.E.; Caruntu, I.-D.; Profire, L. Recent Biomedical Approaches for Chitosan Based Materials as Drug Delivery Nanocarriers. Pharmaceutics 2021, 13, 587
  15. Venkatesan, J.; Kim, S.-K. Chitosan Composites for Bone Tissue Engineering—An Overview. Mar. Drugs 2010, 8, 2252-2266.
  16. Reves, Benjamin T., Jessica A. Jennings, Joel D. Bumgardner, and Warren O. Haggard. 2011. "Osteoinductivity Assessment of BMP-2 Loaded Composite Chitosan-Nano-Hydroxyapatite Scaffolds in a Rat Muscle Pouch" Materials 4, no. 8: 1360-1374
  17. Zimoch-Korzycka, A.; Śmieszek, A.; Jarmoluk, A.; Nowak, U.; Marycz, K. Potential Biomedical Application of Enzymatically Treated Alginate/Chitosan Hydrosols in Sponges—Biocompatible Scaffolds Inducing Chondrogenic Differentiation of Human Adipose Derived Multipotent Stromal Cells. Polymers 2016, 8, 320.
  18. Sampath, U.G.T.M.; Ching, Y.C.; Chuah, C.H.; Sabariah, J.J.; Lin, P.-C. Fabrication of Porous Materials from Natural/Synthetic Biopolymers and Their Composites. Materials 2016, 9, 991.
  19. ] Paun, I.A.; Popescu, R.C.; Calin, B.S.; Mustaciosu, C.C.; Dinescu, M.; Luculescu, C.R. 3D Biomimetic Magnetic Structures for Static Magnetic Field Stimulation of Osteogenesis. Int. J. Mol. Sci. 2018, 19, 495.
  20. Bealer, E.J.; Kavetsky, K.; Dutko, S.; Lofland, S.; Hu, X. Protein and Polysaccharide-Based Magnetic Composite Materials for Medical Applications. Int. J. Mol. Sci. 2020, 21, 186.
  21. de Menezes, F.L.; Andrade Neto, D.M.; Rodrigues, M.d.L.L.; Lima, H.L.S.; Paiva, D.V.M.; da Silva, M.A.S.; Fechine, L.M.U.D.; Sombra, A.S.B.; Freire, R.M.; Denardin, J.C.; Rosa, M.d.F.; de Souza Filho, M.d.S.M.; Mazzetto, S.E.; Fechine, P.B.A. From Magneto-Dielectric Biocomposite Films to Microstrip Antenna Devices. J. Compos. Sci. 2020, 4, 144
  22. .Fiejdasz, S.; Gilarska, A.; Strączek, T.; Nowakowska, M.; Kapusta, C. Magnetic Properties of Collagen–Chitosan Hybrid Materials with Immobilized Superparamagnetic Iron Oxide Nanoparticles (SPIONs). Materials 2021, 14, 7652
  23. Sergi, R.; Bellucci, D.; Cannillo, V. A Review of Bioactive Glass/Natural Polymer Composites: State of the Art. Materials 2020, 13, 5560
  24. Roffi, A.; Kon, E.; Perdisa, F.; Fini, M.; Di Martino, A.; Parrilli, A.; Salamanna, F.; Sandri, M.; Sartori, M.; Sprio, S.; Tampieri, A.; Marcacci, M.; Filardo, G. A Composite Chitosan-Reinforced Scaffold Fails to Provide Osteochondral Regeneration. Int. J. Mol. Sci. 2019, 20, 2227.
  25. Sobczak-Kupiec, A.; Drabczyk, A.; Florkiewicz, W.; Głąb, M.; Kudłacik-Kramarczyk, S.; Słota, D.; Tomala, A.; Tyliszczak, B. Review of the Applications of Biomedical Compositions Containing Hydroxyapatite and Collagen Modified by Bioactive Components. Materials 2021, 14, 2096.
  26. Kaczmarek-Szczepańska, B.; Pin, J.M.; Zasada, L.; Sonne, M.M.; Reiter, R.J.; Slominski, A.T.; Steinbrink, K.; Kleszczyński, K. Assessment of Melatonin-Cultured Collagen/Chitosan Scaffolds Cross-Linked by a Glyoxal Solution as Biomaterials for Wound Healing. Antioxidants 2022, 11, 570
  27. Râpă, M.; Gaidau, C.; Mititelu-Tartau, L.; Berechet, M.-D.; Berbecaru, A.C.; Rosca, I.; Chiriac, A.P.; Matei, E.; Predescu, A.-M.; Predescu, C. Bioactive Collagen Hydrolysate-Chitosan/Essential Oil Electrospun Nanofibers Designed for Medical Wound Dressings. Pharmaceutics 2021, 13, 1939
  28. Castro, J.I.; Valencia-Llano, C.H.; Valencia Zapata, M.E.; Restrepo, Y.J.; Mina Hernandez, J.H.; Navia-Porras, D.P.; Valencia, Y.; Valencia, C.; Grande-Tovar, C.D. Chitosan/Polyvinyl Alcohol/Tea Tree Essential Oil Composite Films for Biomedical Applications. Polymers 2021, 13, 3753.
  29. Yang, Y.; Wang, Z.; Xu, Y.; Xia, J.; Xu, Z.; Zhu, S.; Jin, M. Preparation of Chitosan/Recombinant Human Collagen-Based Photo-Responsive Bioinks for 3D Bioprinting. Gels 2022, 8, 314.
  30. Nogueira, L.F.B.; Eufrásio Cruz, M.A.; Aguilar, G.J.; Tapia-Blácido, D.R.; da Silva Ferreira, M.E.; Maniglia, B.C.; Bottini, M.; Ciancaglini, P.; Ramos, A.P. Synthesis of Antibacterial Hybrid Hydroxyapatite/Collagen/Polysaccharide Bioactive Membranes and Their Effect on Osteoblast Culture. Int. J. Mol. Sci. 2022, 23, 7277
  31. Fang, Q.; Yao,Z.; Feng, L.; Liu, T.; Wei, S.; Xu, P.; Guo, R.; Cheng, B.; Wang, X. Antibiotic-loaded chitosan-gelatin scaffolds for infected seawater immersion wound healing. International Journal of Biological Macromolecules, 2020, 159, 1140-1155.
  32. Sionkowska, A.; Walczak, M.; Michalska-Sionkowska, M. Preparation and characterization of collagen/chitosan composites with silver nanoparticles. Polymer Composites. 2020; 41: 951–957.
  33. Tallapaneni, V.; Pamu,D.; Mude, L.; Dual-Drug Loaded Biomimetic Chitosan-Collagen Hybrid Nanocomposite Scaffolds for Ameliorating Potential Tissue Regeneration in Diabetic Wounds bioRxiv 2022.02.16.480700; DOI: https://doi.org/10.1101/2022.02.16.480700
  34. Tripathi, S.; Singh, B.N.; Singh, D. et al. Optimization and evaluation of ciprofloxacin-loaded collagen/chitosan scaffolds for skin tissue engineering. 3 Biotech, 2020, 11, 160. https://doi.org/10.1007/s13205-020-02567-w
  35. Ioan, D.-C.; Rău, I.; Albu Kaya, M.G.; Radu, N.; Bostan, M.; Zgârian, R.G.; Tihan, G.T.; Dinu-Pîrvu, C.-E.; Lupuliasa, A.; Ghica, M.V. Ciprofloxacin-Collagen-Based Materials with Potential Oral Surgical Applications. Polymers 2020, 12, 1915
  36. Bealer, E.J.; Onissema-Karimu, S.; Rivera-Galletti, A.; Francis, M.; Wilkowski, J.; Salas-de la Cruz, D.; Hu, X. Protein–Polysaccharide Composite Materials: Fabrication and Applications. Polymers 2020, 12, 464.
  37. Yuan, L.; Yao, Q.; Liang, Y. et al. Chitosan based antibacterial composite materials for leather industry: a review. J Leather Sci Eng 2021, 3,12. https://doi.org/10.1186/s42825-020-00045-w
  38. Villanueva-Ornelas G.E.; Nunez-Anita R.E.; Arenas-Arrocena M.C.; Luevano-Colmenero G.H.; Acosta-Torres L.S. Biocidal and bioresorbable chitosan/triclosan/collagen matrixes. Innovation. 2019.2(5):e1. doi:10.30771/2019.
  39. Sánchez-Machado, D.; López-Cervantes, J.; Martínez-Ibarra, D.; Escárcega-Galaz, A; Vega-Cázarez, C. The use of chitosan as a skin-regeneration agent in burns injuries: A review. e-Polymers, 2022, 22(1), 75-86.
  40. Pasaribu, K.M.; Gea, S.; Ilyas, S.; Tamrin, T.; Radecka, I. Characterization of Bacterial Cellulose-Based Wound Dressing in Different Order Impregnation of Chitosan and Collagen. Biomolecules 2020, 10, 1511.
  41. Ferdes, C. Ungureanu, N. Radu, A. Chirvase 2009. Antimicrobial effect of Monascus sp red rice against some bacterial and fungal strains. Chemical Engineering Transaction 17, 1089-1094
  42. Zaharie, M-G.O.; Radu, N.; Pirvu, L.; Bostan, M.; Voicescu, M.; Begea, M.; Constantin, M.; Voaides, C.; Babeanu, N.; Roman, V. Studies Regarding the Pharmaceutical Potential of Derivative Products from Plantain. Plants 2022, 11, 1827.
  43. Radu, N.; Salageanu, A.; Ferdes, M.; Rau, I. 2012. Cytotoxicity Study Regarding Some Products Derived from Monascus sp., Cryst. Liq. Cryst., 555,189-194
  44. Ikeda,T.; Ikeda, K.; Yamamoto,K.; Ishizaki, H.; Yoshizawa, Y.; Yanagiguchi, K.; Yamada, S.; Hayashi, Y. Fabrication and Characteristics of Chitosan Sponge as a Tissue Engineering Scaffold", BioMed Research International, 2014, 2014, 786892.
  45. Horn, M. M.; Amaro Martins,V.C; de Guzzi Plepis A.M. Interaction of anionic collagen with chitosan: Effect on thermal and morphological characteristics. Carbohydrate Polymers, 2009, 77(2), 239-243,
  46. Cuong, H.N.; Minh, N.C.; Van Hoa,N.; Trung,T.S. Preparation and characterization of high purity β-chitin from squid pens (Loligo chenisis). International Journal of Biological Macromolecules, 2016, 93, 442-447.
  47. Sionkowska, A.; Kaczmarek, B.; Stalinska, J.; Osyczka, A. M. Biological Properties of Chitosan/Collagen Composites. In Key Engineering Materials, 2013, 587, 205–210.
  48. Wang, X., Wang, G., Liu, L.et al. The mechanism of a chitosan-collagen composite film used as biomaterial support for MC3T3-E1 cell differentiation. Sci Rep, 2016, 6, 39322.
  49. Udkhiyati, M.; Rosiati, N. M.; Silvianti, F. The Influence of Chitosan Towards Antibacterial Properties in Natural Leather. Leather & Footwear Journal / Revista de Pielarie Incaltaminte, 2020, 20(4), 425–434.
  50. Hua, Y.; Ma, C.; Wei, T.; Zhang, L.; Shen, J. Collagen/Chitosan Complexes: Preparation, Antioxidant Activity, Tyrosinase Inhibition Activity, and Melanin Synthesis. Int. J. Mol. Sci. 2020, 21, 313.
  51. Al-Ghamdi, M.; Aly, M.M.; Sheshtawi, R.M. Antimicrobial Activities of Different Novel Chitosan-Collagen Nanocomposite Films Against Some Bacterial Pathogens. International Journal of Pharmaceutical and Phytopharmacological Research, 2020, 10(1), 114-121.
  52. Li,J.; Zhuang, S. Antibacterial activity of chitosan and its derivatives and their interaction mechanism with bacteria: Current state and perspectives, European Polymer Journal, 2020, 138, 109984.
  53. Ke, C.-L.; Deng, F.-S.; Chuang, C.-Y.; Lin, C.-H. Antimicrobial Actions and Applications of Chitosan. Polymers 2021, 13, 904.
  54. Frosini, S.M.; Bond, R.. Activity In Vitro of Clotrimazole against Canine Methicillin-Resistant and Susceptible Staphylococcus pseudintermedius. Antibiotics (Basel), 2017, 6(4):29.
  55. Grimling, B.; Karolewicz, B.; Nawrot, U.; Włodarczyk, K..; Górniak, A.Physicochemical and Antifungal Properties of Clotrimazole in Combination with High-Molecular Weight Chitosan as a Multifunctional Excipient. Mat Drugs, 2020, 18(12):591.
  56. Martins, A.M.; Marto, J.M.; Johnson, J.L.; Graber, E.M. A Review of Systemic Minocycline Side Effects and Topical Minocycline as a Safer Alternative for Treating Acne and Rosacea. Antibiotics 2021, 10, 757

Request 9. The abstract should be rewritten to reflect the significance of the proposed work. The current abstract shows a lot of background information.

Answer : The Abstract of paper was rewritten as follow:

Raw materials, such as collagen and chitosan, obtained from by-products from the food industry (beef hides, crustacean exoskeletons) can be used to obtain collagen-chitosan composite biomaterials, with potential applications in regenerative medicine. Functionalization of these composite biomaterials is a possibility, thus resulting a molecule with potential applications in regenerative medicine, namely clotrimazole (a molecule with antibacterial, antifungal and antitumor activity), at a mass ratio (collagen:chitosan clotrimazole) of 1:1:0.1. This functionalized composite biomaterial has the greatest potential for application in regenerative medicine, due to the following properties: 1) it is porous and the pores formed are interconnected between them, due to the use of a mass ratio between collagen: chitosan=1:1; 2) the size of the formed pores is between 500-50μm; 3) between collagen and chitosan, hydrogen bonds are formed, which ensure the unity of composite biomaterial; 4) the functionalized bio-composite exhibits in vitro antimicrobial activity for Candida albicans, Staphylococcus aureus and Staphylococcus aureus MRSA; for the latter microorganism, the antimicrobial activity is equivalent to that of the antibiotic Minocycline; 5) the proliferation tests performed on a standardized line of normal human cells with simple or composite materials obtained by lyophilization do not show cytotoxicity in the concentration range studied (10-500) μg/mL.

Request 10. Conclusion: What are the advantages and disadvantages of this study compared to the existing studies in this area?

Answer: This request was solved in the manuscript, as follows:

The novelty of this study consists in obtaining a porous composite biomaterial with interconnected pores, functionalized with clotrimazole, with pore size=(500÷50)μm, for which the studies performed in vitro showed that:

1) it exhibits significant antimicrobial activity for gram-positive microorganisms such as Staphylococcus aureus and, respectively, for antibiotic-resistant microorganisms such as Staphylococcus aureus MRSA;

2) it does not exhibit cytotoxicity;

3) has a potential for application in regenerative medicine, as a result of its own components (collagen, chitosan, clotrimazole) that support tissue regeneration; it inhibits the development of pathogenic microorganisms such as Candida albicans, Staphylococcus aureus or antibiotic-resistant microorganisms such as Staphylococcus aureus MRSA.

Request 11.The space between value and units may be eliminated.

Answer: All these were done in manuscript

Request 12.The numbers for the all equations have to be provided.

Answer: All these were done in manuscript

Request 13.Add future work in bullets.

This request was solved in the manuscript, as follows:

To validate the applicative potential of this functionalized composite biomaterial additional studies are needed:

  • testing the obtained composite biomaterials obtained on different tumor cell lines;
  • carrying out the preclinical tests on the lab animals in order to evaluate the antimicrobial activity ''in vivo'';
  • carrying out preclinical tests on lab animals to confirm the absence of cytotoxicity ''in vivo’’

Reviewer 2 Report

Babeanu et al. present a very interesting manuscript, well structured and with adequate results that support the conclusions. However, there are some points that authors should take into consideration:

-In the first place, the title of the manuscript is too generic and does not represent the nature of the research.

-The summary is too brief and should be modified with more data on the results and conclusions.

-The authors must update the references used in the introduction, as well as introduce a translational justification for the research.

-The methodology section is adequate, but an explanatory figure of the process should be included so that the reader has a global vision of the reproducibility of the study.

-The results are well described, but in general terms the figures are of low quality. The authors must unify the format of the graphs.

-On the other hand, the description of the figures in figure legends should be more extensive.

-This manuscript has enough entity for the authors to carry out an independent discussion in its own section. This manuscript should have a discussion, since it is not just a proof of concept.

-The authors must improve very extensively the use of English grammar.

Author Response

Dear Reviewer,

Thank you for your suggestions (requests)

All of that was used in our manuscript, in the following way

Request 1 In the first place, the title of the manuscript is too generic and does not represent the nature of the research.

Answer: the title of the manuscript was changed as follows:

Obtaining and characterization of composite biomaterials obtained from animal resources with potential applications in regenerative medicine.

Request 2. The summary is too brief and should be modified with more data on the results and conclusions.

Answer: the summary was improved as follows:

Raw materials, such as collagen and chitosan, obtained from by-products from the food industry (beef hides, crustacean exoskeletons) can be used to obtain collagen-chitosan composite biomaterials, with potential applications in regenerative medicine. Functionalization of these composite biomaterials is a possibility, thus resulting a molecule with potential applications in regenerative medicine, namely clotrimazole (a molecule with antibacterial, antifungal and antitumor activity), at a mass ratio (collagen:chitosan clotrimazole) of 1:1:0.1. This functionalized composite biomaterial has the greatest potential for application in regenerative medicine, due to the following properties: 1) it is porous and the pores formed are interconnected between them, due to the use of a mass ratio between collagen: chitosan=1:1; 2) the size of the formed pores is between 500-50μm; 3) between collagen and chitosan, hydrogen bonds are formed, which ensure the unity of composite biomaterial; 4) the functionalized bio-composite exhibits in vitro antimicrobial activity for Candida albicans, Staphylococcus aureus and Staphylococcus aureus MRSA; for the latter microorganism, the antimicrobial activity is equivalent to that of the antibiotic Minocycline; 5) the proliferation tests performed on a standardized line of normal human cells with simple or composite materials obtained by lyophilization do not show cytotoxicity in the concentration range studied (10-500) μg/mL.

Request 3 The authors must update the references used in the introduction, as well as introduce a translational justification for the research.

Answer: the references used were updated (the number of references used was 56), and the objective of the research was introduced as follows:

Regenerative medicine requests new biocompatible formulations with enhanced properties. These types of formulations are obtained currently from cheap raw materials, such as the sub-products resulting from the food industry like animal hides and crustacean exoskeletons (chitin). These sub-products represent the main natural resources for obtaining biopolymers such as hydrolyzed collagen and chitosan. A promising molecule with the possibility to enhance the properties of the biomaterials obtained from chitosan and/or collagen is clotrimazole, an imidazole derivative used in the treatment of fungal infections.

The aim of this study was to obtain and characterize composite biomaterials based on collagen hydrolyzate, chitosan and clotrimazole.

The main objectives of the research were:

1) obtaining collagen:chitosan composite biomaterials in the absence or the presence of an antibiotic reagent, with potential use in regenerative medicine;

2) characterization of the morphology of the composite biomaterials obtained, compared to the biomaterials resulting from raw materials (collagen, chitosan);

3) highlighting the structural characteristics of the obtained biomaterials (highlighting the functional chemical groups, the interactions between molecules, and the triple helix structure from collagen in simple or composite biomaterials ;

4) evaluation of the antimicrobial properties of the simple or composite biomaterials;

5) evaluation of the cytotoxicity of the obtained simple biomaterials/composite biomaterials;

6) selection of composite biomaterial(s) with potential applications in regenerative medicine.

Request 4 The methodology section is adequate, but an explanatory figure of the process should be included so that the reader has a global vision of the reproducibility of the study.

Answer: this request was solved as follows:

The research methodology applied in this research work is presented in figure 2.

Figure 2. Research methodology applied in research studies

Request 5 The results are well described, but in general terms the figures are of low quality. The authors must unify the format of the graphs.

Answer: these requests were solved into the manuscript

Request 6 On the other hand, the description of the figures in figure legends should be more extensive.

Answer: these requests were solved in the manuscript

Request 7 This manuscript has enough entity for the authors to carry out an independent discussion in its own section. This manuscript should have a discussion, since it is not just a proof of concept.

Answer: these requests were solved in the manuscript, by introduction of an independent Chapter entitled ‘’Discussions’’ as follows:

The ESI-MS analysis showed that the collagen hydrolyzate contains species with a molecular mass of 0.3-2.2 KDa, values that are in agreement with reported data for this type of raw material [1-2]. The infrared analysis of the collagen hydrolyzate was used to highlight the existence of a structure of triple helix for collagen [46-47] in this solution. In the case of the chitosan solution made in 0.5M acetic acid, the existence of the monomers with an average mass of (0.645÷3.1) KDa was highlighted, which confirms the existence of chitosan structures type "coil" or ''wrapped coil'' in the acetic solution [7; 10].

               The physical-chemical analyses achieved showed that the biomaterials resulting from lyophilization have a porous structure, the best composite biomaterial being obtained at a mass ratio between collagen and chitosan of  1:1. The composite biomaterial obtained at this mass ratio has a porous structure (pores with sizes between (1000÷100)μm) with interconnected pores, which ensures the circulation of oxygen through the biocomposite material [44-46]. In the biocomposite functionalized with clotrimazole, the average pore size decreases (the diameter of the resulting pores is 500÷50 μm),  compared to the non-functionalized biocomposite, obtained at the same mass ratio.

               The results obtained from the analysis of infrared spectra showed the following:

  • in the collagen hydrolyzate used, collagen had the triple helix structure, the structure that is also preserved in the composite biomaterials obtained by lyophilization, for which the ratio between the two peaks is slightly shifted to the lower wave numbers, (the obtained values for the three composite biomaterials S4, S5 and S6 are: 3; 2.14 and 1.5) [46-47];
  • the theoretical degree of deacetylation of pharmaceutical chitosan calculated with relation (1) is 65% [10]. Most probable, this degree of deacetylation represents the reason for inhomogeneities that appear in the morphology of the chitosan sponge (figure 3d3). These inhomogeneities also appear in the morphology of biomaterial composite of collagen:chitosan:clotrimazole, obtained at a mass ratio of 1:1:0.1 (Figures 3b6,  3e3);
  • in all the three composite biomaterials, the presence of hydrogen bonds between the chitosan and collagen molecules was highlighted.

               The tests carried out in vitro on the most common pathogenic microorganisms from medical practice showed that the biomaterials or composite biomaterials that do not contain clotrimazole have a local antimicrobial activity (inhibition diameters between 7-8 mm), results which are in agreement with other data reported in the literature [30; 33].

               The presence of clotrimazole significantly increases the antimicrobial activity of the biomaterial composite, for fungi such as Candida albicans (for which clotrimazole is recognized to have selective action [54-55]) and, respectively, for gram-positive bacteria such as Staphylococcus aureus and Staphylococcus aureus MRSA.

               For medical applications, the biocomposite functionalized with clotrimazole has the greatest application potential in this field, as the existence of this type of composite biomaterial has not been reported in current literature. 

The tests performed ''in vitro'' on Staphylococcus aureus with the composite biomaterial functionalized with clotrimazole showed that it has greater antimicrobial activity than the 0.1% clotrimazole solution, the biological activity recorded being comparable to that of the antibiotic Tobramycin.

In the case of the tests carried out on Staphylococcus aureus MRSA, the results obtained showed that the antimicrobial activity of the functionalized composite biomaterial is higher than in the case of common antibiotics such as Clotrimazole (0.1 or 1%) and Tobramycin, being comparable to the antibiotic Minocycline, which is used in medical practice for treating/preventing infections with Staphylococcus aureus or Staphylococcus aureus MRSA [56]. Proliferation tests performed on a standardized human cell line (HUVEC) for 72 hours showed that the obtained biomaterials are not cytotoxic, except for the composite biomaterial obtained at a mass ratio = collagen:chitosan = 1:1, for which cell viability drops to 50%, at a concentration of it in the culture medium of 500 μg/mL. This value is most likely due to experimental errors because at 24 and 48 hours the proliferation index is higher than 1. This fact once again confirms the lack of cytotoxicity of biomaterials of this type reported in the literature [26, 29; 34-35].

Request 8 The authors must improve very extensively the use of English grammar.

Answer: this request was solved in the manuscript

Round 2

Reviewer 1 Report

Quality of Figure 4 should be improved. The numbers are not clear.

Main features of Figure 3 must be included in the figures.

Please add a reference to The proliferation index.

The introduction should be supported by:  Biodegradable Magnesium Metal Matrix Composites for Biomedical Implants: Synthesis, Mechanical Performance, and Corrosion Behavior-A Review; In Vitro Degradability, Microstructural Evaluation, and Biocompatibility of Zn-Ti-Cu-Ca-P Alloy

What are NHEK and NHDF?

Nomenclature section should be included.

“-measurements performed after 72h of exposure show that the cell viability was greater than 93% for all samples except the S3 at c=500 μg/mL, for which the viability remains at 57.8.”; please check this sentence.

What are the measurement tools? Specifications should be included.

Author Response

Dear Reviewer,

Thank you for your feedback. All requests were solved into manuscript, as follow:

Request 1 Quality of Figure 4 should be improved. The numbers are not clear.

Answer: All these were done by increased font dimensions, as follow:

Request 2 Main features of Figure 3 must be included in the figures.

Answer; Main features were added in Figures 3,  as follow:

Figure 3a Morphology of collagen sponge obtained by lyophilization. 3a1) total view of collagen sponge, with rare pores (magnification 40x); 3a2) view regarding unregulated pore geometry, more with diameters less than under 1 mm in diameters (magnification 10x); 3a3) dimension of the rare small pore, with diameters of 250 microns (magnification 250x).

Figure 3b Morphology of chitosan sponge obtained by lyophilization: 3b1) lamellar structure of chitosan sponge (magnification 250x); 3b2) lamellas from a sponge of chitosan; distance between lamellas under 200μm (magnification 500x); 3b3) details regarding pores from lamellar structure, with dimensions under 100μm (magnification 1000x); ; 3b4) interconnected pore distribution inside of sponge of chitosan (magnification 40x); 3b5) pores geometry inside of chitosan sponge (magnification 500x); 3b6) pores geometry from sponge of chitosan and inhomogeneities presence in its internal structure (magnification 1000x); 

Figure 3c. Morphology of composite biomaterial between collagen and chitosan, obtained at collagen: chitosan mass ratio of 1:1 by lyophilization: 3c1) porous structure of the collagen-chitosan sponge, with interconnected pores (magnification 100x); 3c2) details regarding interconnected pores geometry (magnification 250x); 3c3) pores geometry, with dimensions of 100μm (magnification 1000x). 

Figure 3d. Morphology of composite biomaterial between collagen and chitosan obtained at collagen: chitosan mass ratio of 3:1. 3d1) porous structure of sponge with pores diameters under 1mm (magnification 100x); 3d2) pores geometry from the sponge with diameters under 500 μm (magnification 250x); 3d3) small pores with dimension under 200 μm microns (magnification 500x). 

Figure 3e. Morphology of sponge composite biomaterial obtained at collagen:chitosan:clotrimazole mass ratio of 1:1:0.1. 3e1) porous structure of composite biomaterial, with pores diameter under 500 microns (magnification 250x); 3e2) details regarding inhomogeneity of internal wall structures and presence of interconnected pores, with diameters under 100μm (magnification 1000x); 3e3) composite biomaterial with pores diameter under 50 μm (magnification 2000x).  

Also the main characteristics of the figures 3 were described in the text as follow:

The studies related to the morphology of the resulting biomaterials showed that the lyophilized products containing collagen (Figures 3a1-3a3) do not contain networks, they have rare pores and the resulting pores have sizes of 1 mm (Figure 3a2) and 500 μm (Figures 3a3). The lyophilized products obtained with collagen cross-linked with acetic acid are stable only at temperatures below 5oC; at room temperature, they melt in 30 minutes (these are not stable at room temperature). In the case of lyophilizates obtained from chitosan cross-linked with acetic acid (chitosan sponges), the images obtained by electron microscopy (Figures 3b1-3b6) show the existence of some lamellar structures (Figures 3b1-3b3), with the lamellae arranged in parallel; distances between the parallel lamellas were < 200 μm (Figures 3b2-3b3). The biomaterial with chitosan has a structure consisting of networks (figure 3b4), with pores arranged along the parallel lamellae. Biomaterial obtained by lyophilization of acetic chitosan solution appears to have inhomogeneities (figures 3b6). The size of the pores in the parallel networks of chitosan was < 100 μm.

The results obtained in the case of sponges of chitosan from this study are comparable to those reported by Ikeda et al. [44], which used chitosan with a high degree of deacetylation (85%) and molecular mass = 100 kDa cross-linked with 2% acetic acid and neutralized with NaOH up to pH 7.4. The resulting lyophilized sponges have a porous structure, with a pore size < 100 μm. The images obtained by electron microscopy for the composite biomaterials obtained at collagen:chitosan=1:1 (mass ratio), show that the sponge obtained has a porous structure (Figures 3c1-3c3), the average pore sizes from its structure being between 500 and 50 μm, (Figures 3c2-3c3), those with a size of 50 μm being predominant. If the mass ratio between collagen and chitosan increases (mass ratio collagen:chitosan = 3:1), then biomaterials with rare pores are obtained (Figure 3d1), with dimensions between 500 μm (Figure 3d2) and 200 μm (Figure 3d3).

The results obtained in this case are similar to those obtained by Horn et al [45], who obtained composite biomaterials from chitosan and collagen; the used chitosan had a low degree of deacetylation (16%) and was obtained from a squid species called Loligo sp. (species of squid, whose exoskeleton contains chitin crystallized in the beta form [46]. The collagen hydrolysate which was used was obtained by alkaline hydrolysis of pig skins, for (24-96h). The analysis of the morphology of the composite biomaterials obtained by lyophilization with hydrolyzed collagen for 96 hours and chitosan from squid, showed that the presence of chitosan has the effect of reducing the size of the pores obtained, the average size of the composite pores being between 56-163μm. The biomaterial functionalized with clotrimazole, which contains collagen, chitosan and clotrimazole in mass ratio = 1:1:0.1, borrows the structure of the composite biomaterial obtained at the same mass ratio but without clotrimazole. The images obtained by electron microscopy indicate a porous structure (Figures 3e1-3e3), with pore sizes between 500 and 50μm (Figures 3e1, 3e3) and an irregular structure in section (Figures 3e2).

Request 3 Please add a reference to The proliferation index.

Answer: Reference was added as follow :

The proliferation index (PI), [42] was calculated according to the following formula (2):

Request 4.The introduction should be supported by:  Biodegradable Magnesium Metal Matrix Composites for Biomedical Implants: Synthesis, Mechanical Performance, and Corrosion Behavior-A Review; n Vitro Degradability, Microstructural Evaluation, and Biocompatibility of Zn-Ti-Cu-Ca-P Alloy

Answer: Introduction was completed as follows

Biomaterials such as the simple chitosan or the composite biomaterials of the chitosan-collagen type have the potential to be used to increase the biocompatibility of some metal alloys, developed for use in the form of implants, such as 1) metal composites based on Zn, Nb, Zr, Co reinforced with magnesium or 2) metal composites containing Zn, Cu, Ti Ca and P [39-40].

The studies carried out at the lab level, on the metal composite based on Zn, Cu, Ti, Ca, and P, have shown good resistance to corrosion of this; additional the determinations made ''in vitro'' with this type of metallic composite showed that its presence favors the proliferation of a Vero cell line, the measured cell viability having values higher than 95% after 24, 48, and 72 h of exposure [40].

The corresponding references were added as follow:

39 Krishnan, R.;  Pandiaraj, S.;  Muthusamy, S.; Panchal, H.; Alsoufi, M.H.; Ibrahim,A.M.M.; Elsheikh,A. Biodegradable magnesium metal matrix composites for biomedical implants: synthesis, mechanical performance, and corrosion behavior – a review,Journal of Materials Research and Technology, 2022, (20), 650-670,

  1. Gopal, N.; Palaniyandi, P.; Ramasamy, P.; Panchal, H.; Ibrahim, A.; Alsoufi, M. S.; & Elsheikh, A. H. In Vitro Degradability, Microstructural Evaluation, and Biocompatibility of Zn-Ti-Cu-Ca-P Alloy. Nanomaterials (Basel, Switzerland), 2022,12(8), 1357.

Request 5. What are NHEK and NHDF?

Request 6.Nomenclature section should be included.

The answers to request 5 and request 6: the explanations regarding NHEK and NHDF are provided in Nomenclature, which  was added at the end of article as follows:

Nomenclature

NHEK = Primary Normal Human Epidermal Keratinocytes (NHEK) represent standardized cell lines, isolated from the epidermis of adult skin.

NHDF =Normal Human Dermal Fibroblasts (NHDF) represent standardized cell lines, isolated from the dermis of adult skin.

Vero cell lines =   represent standardized cell lines, the original Vero cell line was established from the kidney of an Africanmonkey.

HUVEC = primary Human Umbilical Vein Endothelial Cells (HUVEC), which represent standardized cell lines, isolated from the human umbilical vein.

Request 7. “-measurements performed after 72h of exposure show that the cell viability was greater than 93% for all samples except the S3 at c=500 μg/mL, for which the viability remains at 57.8.”; please check this sentence.

Answer: The sentence was revised as follows:

‘’measurements performed after 72h of exposure show that the cell viability was greater than 93% for all samples except the S3 at c=500 μg/mL, for which the viability decreased at 57.8%’’

Request 8.

What are the measurement tools? Specifications should be included.

All mesassurements tools were provided in manuscript, as follow:

  • For Stirring: magnetic stirrer, Heidolph Instruments GmbH Schwabach, Germany;
  • For ESI -MS studies : The determinations were made using a device type LC-MS/TOF 6224 AGILENT Technology Houston, USA).
  • For sublimation process: The sublimation process was carried out under a vacuum and was carried out with a Labconco 77530 lyophilizer (Labconco 77530, Kansas, USA);
  • For SEM studies: The morphology of the composite biomaterials was obtained through electron microscopy studies, carried out with FEI Quanta 200 Scanning Electron Microscope with Ambient Scanning (Thermo Fisher, Waltham, USA), magnifications of over 100 000x, operated at a pressure under 1.3 Pa.
  • For Infrared spectroscopy analysis: The infrared spectra were recorded in the range (400÷400 cm-1), using an FT-IR GX Perkin Elmer spectrophotometer (Massachusetts, USA) equipped with an ATR device, a Dynascan interferometer and a DTGS (deuterated triglycine sulfate) detector.
  • For antimicrobial activities was used a Laminar hood type Faster Bio 48 (Cornaredo, Italy) an incubator Cole Palmer H2200-H-E (Vernon Hills, USA)
  • For proliferation studies were used: a microbiological hood with laminar flow and an Elisa EZ 400 Biochrom analyzer (Holliston, USA)

Round 3

Reviewer 1 Report

Accept.